# Value-Based Abstraction Functions for Abstraction Sampling

**Bobak Pezeshki**[1]        **Kalev Kask**[1]        **Alexander Ihler**[1]        **Rina Dechter**[1]

[1]University of California, Irvine

## Abstract

Monte Carlo methods are powerful tools for solving problems involving complex probability distributions. Despite their versatility, these methods often suffer from inefficiencies, especially when dealing with rare events. As such, importance sampling emerged as a prominent technique for alleviating these challenges. Recently, a new scheme called Abstraction Sampling was developed that incorporated stratification to importance sampling over graphical models. However, existing work only explored a limited set of abstraction functions that guide stratification. This study introduces three new classes of abstraction functions combined with seven distinct partitioning schemes, resulting in twenty-one new abstraction functions, each motivated by theory and intuition from both search and sampling domains. An extensive empirical analysis on over 400 problems compares these new schemes highlighting several well-performing candidates.

## 1 INTRODUCTION

The partition function ($Z$) is an important quantity in probabilistic graphical model inference and is often estimated using Monte Carlo methods such as Importance Sampling (IS) [Rubinstein and Kroese, 2016, Liu et al., 2015, Gogate and Dechter, 2011]. Inspired by the works of Knuth [1975] and Chen [1992], a framework called Abstraction Sampling (AS) [Broka et al., 2018] was introduced extending IS by enabling samples to represent multiple configurations. AS uses concepts from Stratified Sampling [Rubinstein and Kroese, 2016, Rizzo, 2007] and Compact Search [Dechter and Mateescu, 2007, Marinescu and Dechter, 2009] to build a sampled subtree called a *probe* which is then used to compute an estimate. Probes are built level-by-level according to a variable ordering where, at each level, an *abstraction function* groups nodes into *abstract states* from which representative nodes are selected to extend paths in the probe.

Using what are referred to as context-based abstraction functions, Broka et al. [2018] showed competitive performance of AS against IS, Weighted Mini-Bucket Importance Sampling (wMBIS) [Liu et al., 2015, Ihler et al., 2012], and IJGP-SampleSearch (IJGP-ss) [Gogate and Dechter, 2011]. Kask et al. [2020] improved Abstraction Sampling scalability with the AOAS algorithm that more efficiently applied AS to AND/OR search spaces. AOAS showed improved performance, additionally comparing to state-of-the-art scheme Dynamic Importance Sampling (DIS) [Lou et al., 2019].

However, AS development has lacked exploration of diverse and potentially more effective abstraction functions. While Hsiao et al. [2023] proposed using graph neural networks to learn abstraction functions, such methods require learning on a corpus of similar problems before use.

**Contributions.** This work provides a detailed study of new abstraction schemes for AS. We present a new class of abstractions defined by real-valued functions aimed at capturing relevant similarity features between nodes. Three classes of this new framework are introduced and augmented by seven partitioning strategies. A purely randomized scheme is also introduced. An extensive empirical evaluation is performed on over 400 problems, comparing our novel schemes against: each other, the previous relCB and randCB abstraction functions [Broka et al., 2018, Kask et al., 2020], and implicitly against IS, wMBIS, IJGP-ss, and DIS.

Our experiments identify three schemes in particular that perform significantly better than any previous scheme. Our results demonstrate a significant improvement for one of the most competitive sampling schemes, thus also yielding a substantial computational advance for one of the most challenging tasks in probabilistic inference.

## 2 GENERAL BACKGROUND

**Graphical Models.** A *graphical model*, such as a Bayesian or Markov network [Pearl, 1988, Darwiche, 2009, Dechter, 2013], can be defined by a 3-tuple $\mathcal{M} = (\mathbf{X}, \mathbf{D}, \mathbf{F})$,

where $\mathbf{X}$ is a set of variables, and $\mathbf{D}$ is the set of variable domains, and $\mathbf{F}$ is a set of functions such that each function $f_\alpha \in \mathbf{F}$ is defined over a subset of variables $\alpha \subseteq \boldsymbol{X}$ (called the function's scope) capturing local interactions. $\mathcal{M}$ defines a global function, often a factorized probability distribution on $\mathbf{X}$, $P(\mathbf{X}) = \frac{1}{Z} \prod_\alpha f_\alpha(X_\alpha)$, where $Z = \sum_X \prod_\alpha f_\alpha(X_\alpha)$, known as the partition function, is a normalization factor. A *primal graph* $\mathcal{G} = (\mathbf{V}, \mathbf{E})$ of $\mathcal{M}$ associates each variable with a node ($\mathbf{V} = \mathbf{X}$) with edges $e \in \mathbf{E}$ connecting nodes whose variables interact locally, appearing in the scope of the same functions.

**Search Spaces of Graphical Models.** A graphical model can be transformed into a compact AND/OR search space to leverage conditional independence and facilitate efficient search algorithms [Dechter and Mateescu, 2007].

Given a primal graph $\mathcal{G}$, an AND/OR search space is defined relative to a *pseudo tree* $\mathcal{T} = (\mathbf{V}, \mathbf{E}')$, a directed rooted tree that captures conditional independence encoded in the model. A pseudo tree $\mathcal{T}$ is constructed according to a variable ordering such that every arc of $\mathcal{G}$ not in $\mathbf{E}'$ is a back-arc in $\mathcal{T}$. This construction ensures conditional independence of any variable and its descendants from variables found in the other branches of $\mathcal{T}$ given assignments to their common ancestors. The pseudo tree in Figure 1a was constructed using a variable ordering $o = [A, B, C, D]$. The dashed line shows an edge in the primal graph that is missing from $\mathcal{T}$, but that would be a back-arc if it were present. From its structure we see that variables $C$ and $D$ are independent of $B$ given assignment to $A$. Here $A$ is referred to as a *branching variable* since it branches to multiple children.

Guided by a pseudo tree $\mathcal{T}$, an *AND/OR search tree* $T$ has alternating levels of OR nodes corresponding to variables and AND nodes corresponding to possible assignments to those variables. Figure 1 shows an AND/OR search tree and its guiding pseudo tree. Note that in the pseudo tree, variables $B$ and $C$ extend to different branches from $A$. Similarly, in the AND/OR search tree, we see OR nodes $B$ and $C$ extending to different branches under each possible assignment of $A$. An arc into an AND node $n_X$ of variable $X$ has a cost $c(n_X)$ equal to the product of functions $f_\alpha \in \mathbf{F}$ such that the path to $n_X$ fully instantiates all $X' \in \alpha$ and such that $X \in \alpha$ [Dechter and Mateescu, 2007].

**Notation.** Capital letters ($X$) represent variables and small letters ($x$) their values. Boldfaced letters represent a collection. Boldfaced capital letters ($\mathbf{X}$) denote a collection of variables, $|\mathbf{X}|$ its cardinality, $D_{\mathbf{X}}$ their joint domains (all possible configurations of $\mathbf{X}$), and bolded $\boldsymbol{x}$ a particular realization in that joint domain (a particular configuration of $\boldsymbol{X}$).

In the context of search, $n$ is used generally to represent nodes in a search tree. For AND/OR search trees, $n_X$ is used to specifically refer to an AND node associated with variable $X$, and $Y_{n_X}$ the OR node associated with variable

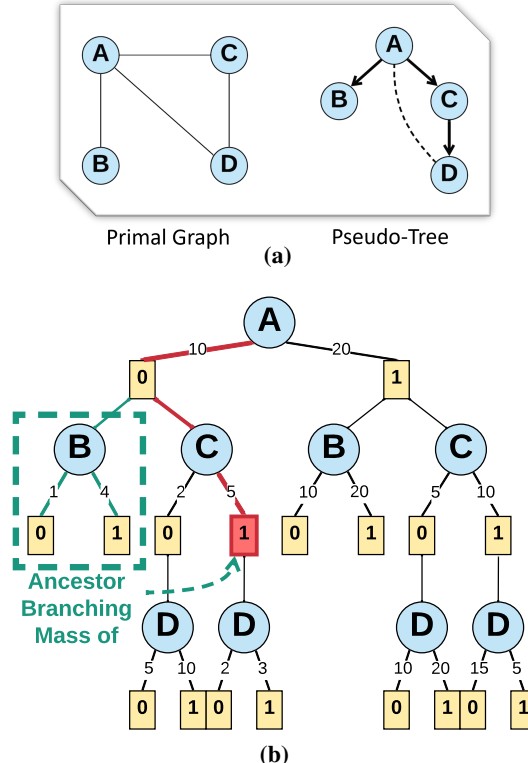

Primal Graph      Pseudo-Tree

**(a)**

**(b)**

**Figure 1:** A full AND/OR tree representing 16 possible full configurations of binary variables $A$, $B$, $C$, and $D$ guided by the pseudo tree shown in subfigure (a) above. The path cost for the highlighted node $n_{A=0,C=1}$ at the end of the path $\rightarrow (A{=}0) \rightarrow (C{=}1)$ is $g(n_{A=0,C=1}) = 10{\cdot}5$. The value of the subtree under $n_{A=0,C=1}$ is $Z(n_{A=0,C=1}) = 2{\cdot}1 + 3{\cdot}1$. Boxed in green is the ancestor branching subtree for $n_{A=0,C=1}$ and it has the value $R(n_{A=0,C=1}) = 1{\cdot}1 + 4{\cdot}1$. Thus, $Q(n_{A=0,C=1}) = (10{\cdot}5){\cdot}(1{\cdot}1 + 4{\cdot}1){\cdot}(2{\cdot}1 + 3{\cdot}1)$.

$Y$ that is the child of $n_X$. $ch(n)$ are the children of node $n$. $path(n)$ is the configuration of the variables along the path from the root of a search tree $T$ to node $n$ according to assignments corresponding to that path. For the highlighted node $n$ in Figure 1b, $path(n) = \{A{=}0, C{=}1\}$. $varpath(n)$ is the set of variables that $path(n)$ provides a configuration for. In Figure 1b $varpath(n) = \{A, C\}$. The cost of the arc to an AND node $n_X$ is

$$c(n_X) = \prod_{f \in \{f_\alpha \in \boldsymbol{F} \,\mid\, \alpha \subseteq varpath(n_X),\, X \in \alpha\}} f(path(n_X)). \quad (1)$$

or 1, vacuously. Letting $anc(n)$ be the AND node ancestors of $n$ in the search tree, the cost of $path(n)$ is $g(n) = \prod_{n' \in anc(n)} c(n')$. In Figure 1b, $g(n) = 10 \cdot 5$.

We now define some important quantities involved in evaluating AND/OR search spaces.

$\boldsymbol{Z(n)}$. The total cost of the subtree rooted at $n$. For an AND node $n_X$ with children OR nodes $Y_{n_X} \in ch(n_X)$, $Z(n_X)$ satisfies

$$Z(n_X) = \prod_{Y_{n_X} \in ch(n_X)} Z(Y_{n_X}) \quad (2)$$

such that for OR nodes $Y_{n_X}$

$$Z(Y_{n_X}) = \sum_{n_Y \in ch(Y_{n_X})} c(n_Y) \cdot Z(n_Y) \qquad (3)$$

with $Z(n_X) = 1$ in the case $n_X$ has no children.

Note that given $n_\varnothing$ as the dummy root node of AND/OR tree $T$, $Z(n_\varnothing) = Z$ of the underlying model $\mathcal{M}$. We denote estimation of $Z(n)$ as $\hat{Z}(n)$. Heuristic estimates of $Z(n)$ are more specifically denoted as $h(n)$.

**$R(n)$.** On the path from the root of an AND/OR tree $T$ to some node $n_X$, there may be an intermediate node $n_Y$ associated with branching variable $Y$ in the guiding pseudo tree $\mathcal{T}$. (In Figure 1b, on the path to the highlighted node $n_{A=0,C=1}$, node $n_{A=0}$ is traversed where $A$ is a branching variable in $\mathcal{T}$ of Figure 1a). When this happens, the remaining variables of the model are split between different branches. Thus, the $Z(n)$ of any node down one of the branches will necessarily omit the costs from the configurations of the variables included in the other branch(es). $R(n_X)$, or the *ancestor branching mass*, captures these omitted costs. (In Figure 1b, the green box shows the portion of $T$ corresponding to $R(n_{A=0,C=1})$.)

More formally, let $br(n_X)$ be the set of ancestor nodes $n_{Y_i}$ of $n_X$ such that each $Y_i$ is a branching variable ancestor of $X$ in $\mathcal{T}$. We then define $R(n_X)$ simply as:

$$R(n_X) = \prod_{n_Y \in br(n_X)} \prod_{\substack{W_{n_Y} \in ch(n_Y) \\ W_{n_Y} \notin path(n_X)}} Z(W_{n_Y}), \qquad (4)$$

(In Figure 1b, $br(n_{A=0,C=1}) = \{n_{A=0}\}$, $A$ being the only branching variable ancestor of $C$ in $\mathcal{T}$, and $B_{n_{A=0}}$ the only respective child OR node ***not*** not on the path to $n_{A=0,C=1}$. Thus, $R(n_{A=0,C=1}) = Z(B_{n_{A=0}}))$. We denote approximations to $R(n)$ as $r(n)$.

**$Q(n)$.** We can now concisely define a quantity $Q(n)$ as the contribution to $Z$ from all full configurations consistent with $path(n)$. In other words, $Q(n)$ is the unnormalized measure of the configuration $path(n)$, with $P(path(n)) = \frac{Q(n)}{Z}$. The quantity $Q(n)$ obeys:

$$Q(n) = g(n) \cdot R(n) \cdot Z(n). \qquad (5)$$

**Example.** In Figure 1b, consider the path from the root to the red node $n_{A=0,C=1}$. Following $n_{A=0}$ to our node, we see OR node $B_{n_{A=0}}$ branches off of the path. So,

$$\begin{aligned} Q(n_{A=0,C=1}) &= g(n_{A=0,C=1}) \cdot R(n_{A=0,C=1}) \cdot Z(n_{A=0,C=1}) \\ &= g(n_{A=0,C=1}) \cdot Z(B_{n_{A=0}}) \quad \cdot Z(n_{A=0,C=1}) \\ &= (10 \cdot 5) \qquad \cdot (1 \cdot 1 + 4 \cdot 1) \quad \cdot (2 \cdot 1 + 3 \cdot 1) \end{aligned}$$

**Stratified Importance Sampling.** Abstraction Sampling builds on Stratified Importance Sampling, which in turn builds on Importance Sampling and Stratified Sampling. *Importance Sampling* is a Monte Carlo scheme used for approximating likelihood queries [Rubinstein and Kroese, 2016, Liu et al., 2015, Gogate and Dechter, 2011]. *Stratified*

---

**Algorithm 1:** AOAS Overview

1. **Initialization:** Begin with a dummy root node $r$.
2. **Probe Generation:** Proceeding in a DFS manner according to a pseudo tree $\mathcal{T}$...
   (a) **Expansion:** Generate children nodes $n$ corresponding to the next variable in the DFS ordering of $\mathcal{T}$. Inherit $w(n)$ from parents and assign appropriate $g(n), h(n),$ and $r(n)$ values.
   (b) **Abstraction:**
      i. **Form Abstract States:** Using $a(\cdot)$, partition newly expanded nodes into abstract states.
      ii. **Select Representative:** Using proposal $p(n) \propto q(n)$, stochastically select a representative from each abstract state and reweigh it such that $w(n) \leftarrow \frac{w(n)}{p(n)}$
   (c) **Backtrack:** After reaching a leaf in $\mathcal{T}$, recursively backtrack until reaching the node that extends to the next unexplored branch of $\mathcal{T}$. While backtracking, update parent node $n''$'s $\hat{Z}(n')$ estimates based on its children's $w(n), g(n),$ and $\hat{Z}(n)$ values.
   (d) **Repeat:** Repeat steps 2a-2c until backtracking to the root node.
3. **Return:** $\hat{Z} = w(r)\,\hat{Z}(r)$ for the root node $r$.

---

*Sampling* is a variance reduction technique for sampling a search space by first dividing it into disjoint strata [Rubinstein and Kroese, 2016]. In *Stratified Importance Sampling*, the sample space is first divided into $k$ strata, then representatives from each strata chosen and re-weighted to represent the omitted members of their respective strata. Rizzo [2007] shows that to reduce overall variance given strata of equal mass under the proposal, the sum of the variances within the strata should be minimized.

## 3 ABSTRACTION SAMPLING

*Abstraction Sampling* (AS) [Broka et al., 2018] applies concepts of Stratified Importance Sampling to sampling over probabilistic graphical models. AS is guided by an abstraction function $a(\cdot)$ that dictates how nodes are partitioned into *abstract states* (abstract states being analogous to strata in stratified sampling). A search tree is iteratively expanded along a variable ordering. After each expansion, $a(\cdot)$ is used to group nodes into abstract states. Then AS uses an importance-sampling-like process to select a representative from each abstract state and reweights it using importance sampling weights to account for the unselected nodes it represents. The selected nodes are then further expanded and the process iterates. This process yields a weighted sampled subtree of the full search tree $T$ as a sample, referred to as a *probe*. It is important to note that AS probes can contain multiple full configurations, whereas samples from importance sampling are each only a single full configuration.

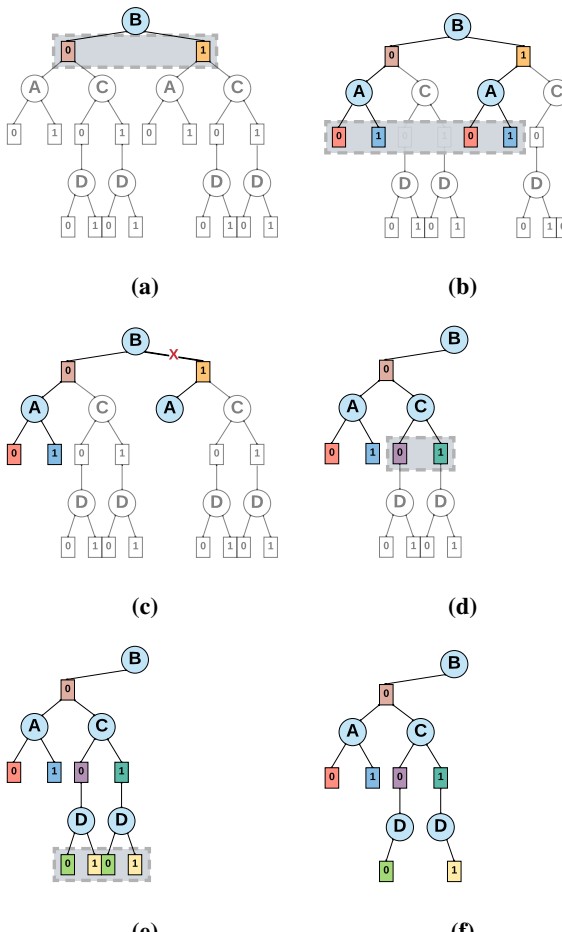

**(a)**       **(b)**

**(c)**       **(d)**

**(e)**       **(f)**

**Figure 2:** From Kask et al. [2020], a sample trace of AOAS following ordering $B \rightarrow A \rightarrow C \rightarrow D$. Transparent nodes indicate portions of the reachable search space yet to be explored. Gray boxes indicate nodes considered for abstraction. Nodes with the same domain values (also indicated by the same color) are abstracted into the same abstract state. Only one node of each color is stochastically selected as a representative for its respective abstract state. Step (c) shows an optional pruning step. Step (f) shows the final probe capturing four full configurations: $B=0, A=0, C=0, D=0$, $B=0, A=1, C=0, D=0$, $B=0, A=0, C=1, D=1$, $B=0, A=1, C=1, D=1$.

**AOAS.** Taking Abstraction Sampling further, Kask et al. [2020] introduced algorithm AOAS that more effectively applied Abstraction Sampling to AND/OR search spaces and significantly improved its performance. AOAS uses a proposal function $p(n) \propto w(n)q(n) = w(n)g(n)h(n)r(n)$ where a weight $w(n)$ accounts for the nodes previously abstracted into the path to $n$, $g(n)$ is the cost of the path to $n$, $h(n)$ is a heuristic estimate of $Z(n)$, and $r(n)$ is an estimate of $R(n)$ (see Figure 3). Algorithm 1 provides a high level description of the AOAS procedure. Figure 2 shows a sample trace of AOAS from Kask et al. [2020]. A more detailed version of the algorithm and detailed description of the sample trace can be found in the Supplemental Materials.

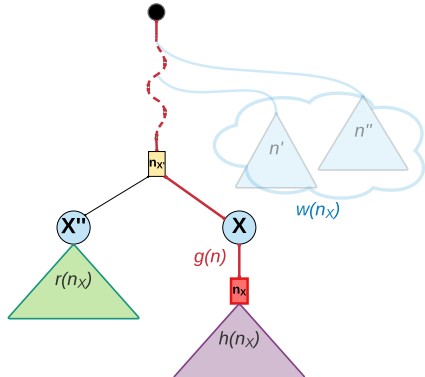

**Figure 3:** The unnormalized proposal distribution $w(n)q(n)$ visualized to show it considering nodes previously abstracted (via $w(n)$), the ancestor branching mass (via $r(n)$), current path cost (via $g(n)$), and subtree mass (via $h(n)$).

## 4 VALUE-BASED ABSTRACTIONS

The choice of abstraction function is a crucial aspect of Abstraction Sampling but has only received limited attention so far. The main focus of this work is to identify new abstraction functions that significantly improve AS performance.

**Existing State-of-the-Art: Context-Based Abstraction Functions.** Broka et al. [2018] designed abstractions based on assignments to a variable's context $C(X)$ - a subset of its ancestors in $\mathcal{T}$ whose assignments uniquely determine the AND/OR subtree below it [Dechter and Mateescu, 2007]. However, the number of configurations to a context is exponential in the context's size. Thus, Broka et al. [2018] and Kask et al. [2020] used *relaxed* context-based (*RelCB*) and *randomized* context-based (*RandCB*) abstractions to control the number of abstract states. RelCB uses a parameter $nCtx$ that groups nodes with the same configuration over the most recent $nCtx-1$ context variables (the relaxed context) into the same abstract state. With a domain size of $k$, this yields at most $k^{nCtx}$ abstract states at each level. RandCB considers the entire context but bounds the number of abstract states per level based on an $nAbs$ parameter and by using a randomized hashing scheme to associate each full context assignment to one of the $nAbs$ abstract states.

**Value-Based Abstractions.** We now introduce a new way to form abstractions that we call Value-Based Abstractions. They are defined by (1) a positive real-valued function $\mu : D_{\boldsymbol{X}} \rightarrow \mathbb{R}^+$, where $D_{\boldsymbol{X}}$ is a set of configurations for the variables $\boldsymbol{X}$, and by (2) a partitioning scheme $\psi_\mu$ that assigns nodes to abstract states based on their $\mu$ value and in an order-consistent manner as defined next.

**Definition 4.1** (Value-Ordered Partitioning)
*Given $nAbs$ and a function $\mu : D_{\boldsymbol{X}} \rightarrow \mathbb{R}^+$, a partitioning function $\psi_\mu : D_{\boldsymbol{X}} \rightarrow \{A_1, A_2, ...A_{nAbs}\}$, is order-consistent with $\mu$ relative to the $nAbs$ abstract states if for any $n_1 \in \boldsymbol{A_i}$ and $n_2 \in \boldsymbol{A_j}$, $i < j \Leftrightarrow \mu(n_1) \leq \mu(n_2)$.*

## 4.1 VALUE-BASED ABSTRACTION CLASSES

We introduce three Value-Based Abstraction classes, each characterized by a unique value function $\mu$ that signifies a notion of similarity between nodes. We will subsequently provide partitioning schemes that, together with $\mu$, will yield a set of full abstraction functions.

**1. Heuristic-Based Abstractions.** Heuristic-Based (HB) abstractions use $\mu(n) = h(n)$, where $h(n)$ is a heuristic estimate of $Z(n)$. Unlike partial or hashed contexts as used by Broka et al. [2018], heuristic estimates of $Z(n)$ can often provide *quantitative* insight into potential similarities of $Z(n)$ values. In particular, this intuition holds when using heuristics that provide bounds on $Z(n)$ such as those via Weighted Mini-Bucket Elimination (wMBE) [Dechter and Rish, 2003, Liu and Ihler, 2011].

**2. Heuristic and Ancestral Branching-Based Abstractions.** Recall that $r(n)$ is an estimate of $n$'s ancestor branching mass $R(n)$. We can show that:

**Theorem 4.1** (AOAS Exact Abstractions)
*If an abstraction function $a(\cdot)$ forms abstract states $\boldsymbol{A_i} \in \boldsymbol{A}$ such that $\exists c_i \in \mathbb{R}^+, \forall n \in \boldsymbol{A_i}, \frac{h(n)r(n)}{Z(n)R(n)} = c_i$ whenever $Z(n)R(n) > 0$ (or $h(n)r(n) = 0$ otherwise), then AOAS is exact with its estimates having zero variance.*

This observation suggests to use $\mathfrak{hr}(n) = \frac{h(n)r(n)}{Z(n)R(n)}$ as a similarity measure. When nodes having close $\mathfrak{hr}$ values are placed in the same abstract state it can lead to a reduction in variance of the resulting estimate. However, without access to $Z(n)$ or $R(n)$ we cannot evaluate this ratio directly. Instead we use the intuition that grouping based on $h(n)r(n)$ may result in sets of nodes also with similar $Z(n)R(n)$, and thus result in similar $\mathfrak{hr}(n)$. We call such schemes that use $\mu(n) = h(n)r(n)$ HR-Based (HRB) abstractions.

**3. Q-Based Abstractions.** Another intuition for generating abstractions comes from statistics theory. In his work on stratified Importance Sampling, Rizzo [2007] showed the potential of overall variance reduction by forming strata (abstract states) having equal mass under the proposal distribution and that minimizes the variance within each strata. Thus, since our proposal $p$ is proportional to $w(n)q(n)$, we use $\mu(n) = w(n)q(n) = w(n)g(n)h(n)r(n)$ in what are called Q-based (QB) abstractions.

## 4.2 ORDERED PARTITIONING SCHEMES

Next we describe seven partitioning schemes $\psi$ to be used with $\mu$ to partition the nodes $\boldsymbol{n}$ into abstract states. Together, $\mu$ and $\psi$ define a value-based abstraction function.

**Running Example.** We will use a running example to illustrate the result of using various partitioning schemes.

Assume we have eight nodes with the following $\mu(n)$:
$$1.0, 1.1, 1.2, 1.3, 1.4, 1.5, 10, 100 \qquad (6)$$

---

**Algorithm 2:** $\Psi_{simpleVB}$

1   $baseCardinality \leftarrow \lfloor \frac{|\boldsymbol{n}|}{nAbs} \rfloor$
2   $extras \leftarrow |\boldsymbol{n}| \mod nAbs$
3   $\boldsymbol{n^*} \leftarrow SORT(\boldsymbol{n}, \mu, \text{low-to-high})$
4   $j_{begin} \leftarrow 1$
5   **foreach** $i \leftarrow 1, ..., nAbs$ **do**
6     **if** $extras > 0$ **then**
7       $j_{end} \leftarrow j_{begin} + baseCardinality$
8       $extras \leftarrow extras - 1$
9     **else**
10      $j_{end} \leftarrow j_{begin} + baseCardinality - 1$
11     $\boldsymbol{A_i} \leftarrow \{n^*_{j_{begin}}, ..., n^*_{j_{end}}\}$
12     $j_{begin} \leftarrow j_{end} + 1$
13   **end**
14   $\boldsymbol{A} \leftarrow \cup_{i=1}^{nAbs} \boldsymbol{A_i}$
15   **return** $\boldsymbol{A}$

---

and want to partition the nodes into $nAbs = 4$ abstract states. As we describe each partitioning scheme, we also demonstrate how the scheme would partition these nodes.

**1. SimpleVB.** The *simpleVB* (simple value-based) scheme groups nodes having similar $\mu(n)$ into the same state by a simple 2-step process: 1) nodes are ordered by $\mu(n)$ (low to high), and 2) nodes are partitioned into abstract states with [approximately] equal cardinality.

*Running Example:* {1.0, 1.1}, {1.2, 1.3}, {1.4, 1.5}, {10, 100}.

This method leverages speed while still aiming to roughly group nodes with similar $\mu(n)$ together.

**2. minVarVB.** *minVarVB* uses Ward's Minimum Variance Hierarchical Clustering, also known as Ward's Method [Ward, 1963] (Algorithm 3), to cluster nodes into $nAbs$ abstract states. Use of Ward's method minimizes total within variance of $\mu(\cdot)$ across all abstract states. Ward's Method can be combined with Lance-Williams linear distance updates [Lance and Williams, 1967] to increase efficiency. More details on Ward's Method and Lance-Williams linear distance updates are found in the Supplemental Materials.

*Running Example:* {1.0, 1.1, 1.2}, {1.3, 1.4, 1.5}, {10}, {100}.

In contrast to *simpleVB*, *minVarVB* places considerable computational resources into computing abstractions by using Ward's Method. Thus *minVarVB* leads to fewer probes being generated but provably forms abstractions that minimize the total within variance of $\mu(n)$ among the abstract states.

**3. equalDistVB.** In attempt to combine the intuition from *minVarVB* and the speed of *simpleVB*, *equalDistVB* greedily adds nodes in order of $\mu$ (low to high) into an abstract state $\boldsymbol{A_i}$ until

$$\mu(\boldsymbol{A_{1,...,i}}) = \sum_{j=1}^{i} \sum_{n \in \boldsymbol{A_j}} \mu(n) \geq \mathcal{Q}_i = \frac{i \cdot \sum_{n' \in \boldsymbol{n}} \mu(n')}{nAbs}, \quad (7)$$

i.e., until the total sum of node values from $\boldsymbol{A_1}, ..., \boldsymbol{A_i}$ reaches or exceeds $\frac{i}{nAbs}$ of the total across all of the nodes

**Algorithm 3:** Ward's Method

1. **Initialization:** Treat each data point as an individual cluster. Assign each cluster a label.

2. **Compute Pairwise Distances:** Calculate the pairwise distances between all clusters. Various distance metrics can be used, such as Euclidean distance.

3. **Cluster Merging Iteration:**

    (a) Identify the pair of clusters $C_i$ and $C_j$ that, when merged into a new cluster $C_{ij}$, results in the smallest increase in the overall within-cluster variance. This is determined using the formula:

    $$\Delta Var = Var(C_{ij}) - (Var(C_i) + Var(C_j))$$

    where $Var(C_{ij})$ is the variance of the merged cluster, and $Var(C_i)$ and $Var(C_j)$ are the variances of clusters $C_i$ and $C_j$, respectively.

    (b) Update distance measures between the newly merged cluster and all other clusters.

4. **Repeat:** Repeat steps 2-3 until the desired number of clusters is achieved.

---

being partitioned. When paired with Q-based abstractions, *equalDistVB* aims to partition nodes into equal mass states under the proposal, motivated by Rizzo [2007].

*Running Example:* {1.0, 1.1, 1.2, 1.3, 1.4, 1.5, 10, 100},{},{},{}.

Although *equalDistVB* hopes to strike a balance between efficiency and low variance of $\mu(n)$ within each abstract state, from the running example we can see it may yield undesirable partitionings for skewed distributions of $\mu(\cdot)$ values. In the example, all of the nodes need to be placed into the first of four abstract states before the sum of their values reaches/exceeds $\frac{1}{4}$ of the total of all nodes being partitioned. Thus, the remaining abstract states end up empty.

**4. equalDistVB2.** A second version of the equalDist scheme, *equalDistVB2*, follows the same general strategy as *equalDistVB* but uses a reversed sort ordering in attempt to mitigate overfilling of abstract states. Modifying the sort order from low-to-high to high-to-low in Line 1 of Algorithm 4 converts *equalDistVB* to *equalDistVB2*.

*Running Example:* {100}, {}, {}, {10, 1.5, 1.4, 1.3, 1.2, 1.1, 1.0}

We see that *equalDistVB2* can still over-pack abstract states. The next two variants aim to mitigate this issue further.

**5. equalDistVB3.** In order to further lessen over-packing and ensure abstract states are not left empty, *equalDistVB3* modifies *equalDistVB2* so that, after processing each abstract state, the next state always has a node added to it by default before checking the abstract state fill condition. Modifying the sort order from low-to-high to high-to-low in Line 1 and $A_i \leftarrow \{\}$ to $A_i \leftarrow \{n_j^*\}; j \leftarrow j + 1;$ in Line 4 of Algorithm 4 converts *equalDistVB* to *equalDistVB3*.

*Running Example:* {100}, {10}, {1.5}, {1.4, 1.3, 1.2, 1.1, 1.0}.

---

**Algorithm 4:** $\Psi_{equalDistVB}$

1   $n^* \leftarrow SORT(n, \mu, \text{low-to-high})$
2   $j \leftarrow 1$
3   **foreach** $i \leftarrow 1, ..., nAbs$ **do**
4     $A_i \leftarrow \{\}$
5     **while** $\mu(A_{1,...,i}) < Q_i$ **do**
6       $A_i \leftarrow A_i \cup \{n_j^*\}$
7       $j \leftarrow j + 1$
8     **end**
9   **end**
10   $A \leftarrow \cup_{i=1}^{nAbs} A_i$
11   **return** $A$

---

While still very efficient, *equalDistVB3* ensures that the provided $nAbs$ granularity is honored, allowing users better control of the search vs. sampling interpolation possible with Abstraction Sampling.

**6. equalDistVB4.** The final equalDist variant, *equalDistVB4*, aims for more even partitioning. Before processing each abstract state $A_i$, a new cut-off is determined based the remaining nodes $n_{rm}^*$ and remaining abstract states:

$$\widehat{Q}_i = \frac{\sum_{n \in n_{rm}^*} \mu(n)}{nAbs - i + 1}. \tag{8}$$

Nodes are added to abstract state $A_i$ while $\mu(A_i) < \widehat{Q}_i$. Modifying the sort order from low-to-high to high-to-low in Line 1 and $\mu(A_{1,...,i}) < Q_i$ to $\mu(A_i) < \widehat{Q}_i$ in Line 5 of Algorithm 4 converts *equalDistVB* to *equalDistVB4*.

*Running Example:* {100}, {10}, {1.5, 1.4, 1.3}, {1.2, 1.1, 1.0}.

Still computationally efficient, *equalDistVB4* spreads nodes with small values more evenly across abstract states.

**7. randVB.** It can be beneficial to rely on randomness to ensure a diverse sampling of abstractions. *randVB* does this by sampling $nAbs - 1$ partition points uniformly at random and without replacement from between nodes sorted according to $\mu(\cdot)$, and then partitions the nodes accordingly. The resulting abstract states ensure that nodes are still grouped according to $\mu(\cdot)$, but the sizes of those groups vary.

---

**Algorithm 5:** $\Psi_{randVB}$

1   $s \sim Unif(\{M \subseteq \{1, ..., |n| - 1\} \mid |M| = nAbs - 1\})$
2   $s^* \leftarrow SORT(s)$
3   $n^* \leftarrow SORT(n, \mu, \text{high-to-low})$
4   $j \leftarrow 1$
5   **foreach** $i \leftarrow 1, ..., nAbs - 1$ **do**
6     $A_i \leftarrow \{n_j^*, ..., n_{s_i^*}^*\}$
7     $j \leftarrow s_i^* + 1$
8   **end**
9   $A_{nAbs} = \{n_j^*, ..., n_{|n^*|}^*\}$
10   $A \leftarrow \cup_{i=1}^{nAbs} A_i$
11   **return** $A$

---

*Running Example:* ex1: {100, 10}, {1.5}, {1.4, 1.3, 1.2}, {1.1, 1.0}; ex2: {100}, {10, 1.5, 1.4, 1.3}, {1.2, 1.1}, {1.0}; etc.

**Complexity.** Assuming $\mu(\cdot)$ is $\mathcal{O}(1)$, each of the proposed partitioning schemes have time complexity $\mathcal{O}(|\boldsymbol{n}| \log |\boldsymbol{n}|)$ and space complexity $\mathcal{O}(|\boldsymbol{n}|)$, with the exception of *minVarVB*, which requires $\mathcal{O}(|\boldsymbol{n}|^2)$ for both. More details can be found in the Supplemental Materials.

## 5 RANDOM-ONLY ABSTRACTIONS

Another unexplored approach was to use purely randomized abstraction schemes. At first glance, one may not expect such schemes to perform well, but randomization in concert with an informative heuristic and proposal can be beneficial.

**Intuition.** First, given an informative heuristic, the stochastic selection of a representative node *within* each abstract state using a good proposal function will typically opt for nodes that represent greater mass, which is generally beneficial in importance sampling. Second, the randomness of node assignments to the abstract states enables nodes that may otherwise have little chance of being selected to occasionally have a greater chance of selection, leading to a more diverse distribution of probes.

**The simpleRand Scheme.** More concisely referred to as RAND, the simpleRand scheme partitions nodes via a 2-step process: 1) nodes first are shuffled to create a uniformly random permutation, and then 2) the nodes are partitioned into (approximately) equal cardinality $nAbs$ abstract states.

---

**Algorithm 6:** $\Psi_{simpleRand}$

1   $baseCardinality \leftarrow \lfloor \frac{|\boldsymbol{n}|}{nAbs} \rfloor$
2   $extras \leftarrow |\boldsymbol{n}| \mod nAbs$
3   $\boldsymbol{n^*} \leftarrow RANDOM\_SHUFFLE(\boldsymbol{n})$
4   $j_{begin} \leftarrow 1$
5   **foreach** $i \leftarrow 1, ..., nAbs$ **do**
6     **if** $extras > 0$ **then**
7       $j_{end} \leftarrow j_{begin} + baseCardinality$
8       $extras \leftarrow extras - 1$
9     **else**
10      $j_{end} \leftarrow j_{begin} + baseCardinality - 1$
11     $\boldsymbol{A_i} \leftarrow \{n^*_{j_{begin}}, ..., n^*_{j_{end}}\}$
12     $j_{begin} \leftarrow j_{end} + 1$
13   **end**
14   $\boldsymbol{A} \leftarrow \cup_{i=1}^{nAbs} \boldsymbol{A_i}$
15   **return** $\boldsymbol{A}$

---

*Running Example:* {1.4, 1.1}, {1.2, 10}, {1.0, 1.3}, {100, 1.5}.

**Complexity.** Both time and space are $\mathcal{O}(|\boldsymbol{n}|)$.

## 6 EMPIRICAL EVALUATION

**Overview.** All combinations of Value-Based Abstraction Classes: Heuristic-Based (**HB**), HR-Based (**HRB**), and Q-Based (**QB**); with each of the Ordered Partitioning Schemes: *simpleVB*, *minVarVB*, *equalDistVB1-4*, and *randVB*; were

tested, resulting in twenty-one value-based abstraction functions. The formerly evaluated context-based (**CTX**) abstraction functions: randCB and relCB were compared against. In addition, the purely random abstraction function, **RAND**, was also included. With the exception of RelCB, each abstraction function uses a hyper parameter, $nAbs$, which bounds the number of abstract states at any level. RelCB instead uses an $nCtx$ parameter that limits the number of context variables used in assigning abstract states. To facilitate comparison, we report RelCB's $nCtx$ parameter instead as an equivalent $nAbs$ parameter assuming a domain size of 2. (For example, if RelCB was run using $nCtx = 6$, we report it with $nAbs = 2^6$). All abstraction functions were tested using the AOAS algorithm [Kask et al., 2020]. All algorithms were implemented in C++. All experiments were run on a 2.66 GHz processor and allotted 8 GB of memory.

**Heuristics.** To inform the sampling proposal, Weighted Mini-Bucket Elimination (wMBE) [Dechter and Rish, 2003, Liu and Ihler, 2011] – which pairs well with AND/OR search [Mateescu and Dechter, 2005] – is used as a heuristic. The i-bound (**iB**) parameter controls the strength of wMBE, where higher i-bounds generally lead to stronger heuristics, and thus better proposals, at the expense of more computation and memory. We standardize our experiments by using the same i-bound when comparing across algorithms.

**Benchmarks.** In line with previous work on Abstraction Sampling, we perform experiments on the same set of over 400 problems from five benchmarks: DBN, Grids, Linkage-Type4, Pedigree, and Promedas used by Kask et al. [2020]. We refer to problem instances with known $Z$ values as *Exact*. Larger problems without exact solutions are called *LARGE*. For LARGE problems, estimates from 10hr of AOAS using the RAND - RAND being well performing - are used as the reference $Z$ value. When experimenting

**Table 1: Exact Benchmark Statistics**. Average statistics for Exact problems. **N**: number of instances, **|X|**: average number of variables, **k**: average of problems' largest domain sizes, **w***: average induced tree-width, **d**: average $\mathcal{T}$ depth.

| Benchmark | N | |X| | k | w* | d |
|---|---|---|---|---|---|
| DBN | 66 | 67 | 2 | 29 | 30 |
| Grids | 8 | 250 | 2 | 22 | 49 |
| Pedigree | 25 | 690 | 5 | 25 | 89 |
| Promedas | 65 | 612 | 2 | 21 | 62 |

**Table 2: LARGE Benchmark Statistics**. Average statistics for LARGE problems. **N**: number of instances, **|X|**: average number of variables, **k**: average of problems' largest domain sizes, **w***: average induced tree-width, **d**: average $\mathcal{T}$ depth.

| Benchmark | N | |X| | k | w* | d |
|---|---|---|---|---|---|
| DBN | 48 | 216 | 2 | 78 | 78 |
| Grids | 19 | 3432 | 2 | 117 | 220 |
| Linkage-Type4 | 82 | 6550 | 5 | 45 | 761 |
| Promedas | 173 | 1194 | 2 | 72 | 114 |

on Exact problems, algorithms use a small i-bound of 5 (weakening the heuristic estimates) and were given a limited time of 300sec to increase difficulty. For LARGE problems, an i-bound of 10 and time limit of 1200sec are used.

**Performance Measure.** To evaluate performance, we define error as: $Error = |\log_{10} \hat{Z} - \log_{10} Z^*|$, where $\hat{Z}$ is the estimate obtained from AS and $Z^*$ is the reference $Z$ value. For Exact problems, $Z^* = Z$.

## 6.1 RESULTS

**Summary Comparison.** To examine potential of the different methods, we tested each algorithm with a range of $nAbs \in \{1, 4, 16, 64, 256, 512, 1024, 2048\}$. For each $nAbs$ and benchmark, we calculated the average error across problems of the benchmark and identified the $nAbs$ that resulted in the lowest average error. Table 3a focuses on Exact problems and shows this lowest average error and corresponding $nAbs$ for each algorithm. Table 3b shows the corresponding results for LARGE problems on the better performing QB and RAND classes, and the CTX class for comparison. If an algorithm was unable to produce a positive Monte Carlo $Z$ estimate for a problem (denoted "Fail"), the wMBE heuristic bound was used as its $Z$ estimate and error computed accordingly. We highlight the best performing schemes.

**Comparison using 100 Samples.** To assess the quality of abstraction functions in an implementation-agnostic manner and irrespective of resulting probe-sizes or speed, we conducted experiments using a one-hundred sample limit (**m-100**). Table 4 shows these results on Exact problems for the better performing QB and Rand classes using $nAbs = 256$. $nAbs = 256$ was chosen as (1) it is an intermediate granularity and (2) all schemes produced 100 samples in a reasonable time. We highlight the best performing schemes.

**Varying nAbs.** Table 5 shows average error for $nAbs \in \{4, 64, 1024\}$ on Exact problems of each benchmark. We focus on the better performing variants of QB: minVarQB, equalDistQB3, equalDistQB4; the purely randomized scheme RAND; and the context-based schemes (CTX) for comparison. In Figure 4 and Figure 5, we also show average error across a wider array of $nAbs$ for minVarQB and equalDistQB4, respectively, the latter also acting as a representative for the profile of equalDistQB3 and RAND.

**Time Series Plot.** Figure 6 and Figure 7 show time-series results for the better performing QB algorithms, RAND, and CTX schemes on a representative Grids and Promedas problem. Each algorithm was plotted with the $nAbs$ that resulted in the lowest average error for the respective benchmark.

## 6.2 ANALYSIS

**Comparison with Context-Based Schemes.** Table 3a shows that there is always a partitioning scheme for HB and HRB that can outperform the best CTX scheme on Exact

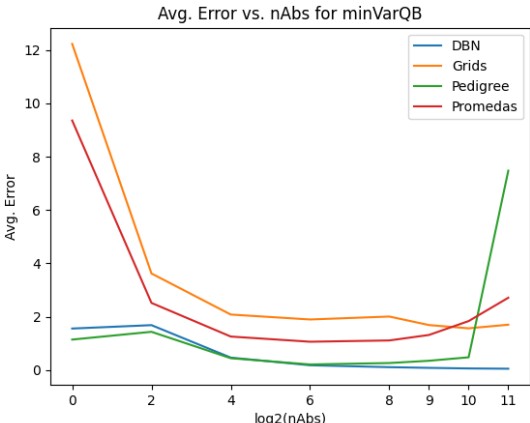

**Figure 4: Varying $nAbs$ for minVarQB**. Average error on Exact problems using iB-5 and time limit 300 sec for each benchmark at various abstraction granularities (in $\log_2$).

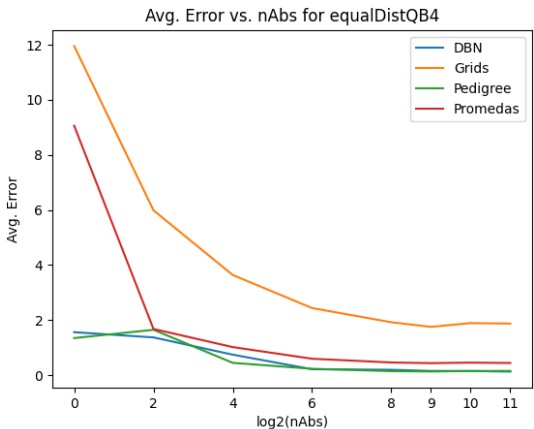

**Figure 5: Varying $nAbs$ for equalDistQB4**. Average error on Exact problems using iB-5 and time limit 300 sec for each benchmark at various abstraction granularities (in $\log_2$).

problems. For HB, the *simple* and *rand* partitioning schemes perform best, whereas for the HRB class it is more benchmark dependent. QB with *minVar*, *equalDist3*, and *equalDist4* partitioning outperform the CTX schemes across all benchmarks. RAND also consistently outperforms the CTX schemes. Results from Table 3b on LARGE problems agree, with the exception of QB with *minVar* and RAND, which fall slightly shy of randCB's performance on Promedas.

**Comparison with Purely Randomized Abstractions.** Table 3 shows RAND is a particularly well performing scheme across all benchmarks. However, the QB class using the *equalDist3* and *equalDist4* strategies is consistently comparable or better than the purely randomized scheme. No other scheme does as well.

**Comparison with Non Abstraction Sampling Schemes.** In prior work by Broka et al. [2018] and Kask et al. [2020], Abstraction Sampling using CTX based abstractions was shown as competitive against several powerful schemes such as Importance Sampling (IS), Weighted Mini-Bucket Importance Sampling (wMBIS) [Liu et al., 2015], IJGP-

**Table (a): iB-5, t-300sec, Exact**

| Class | Scheme | DBN nAbs | Fail | Avg. Error | Grids nAbs | Fail | Avg. Error | Pedigree nAbs | Fail | Avg. Error | Promedas nAbs | Fail | Avg. Error |
|---|---|---|---|---|---|---|---|---|---|---|---|---|---|
| HB | simple | 2048 | 0 | 0.440 | 1024 | 0 | 2.202 | 2048 | 0 | 0.150 | 1024 | 0 | 0.575 |
| | minVar | 1 | 0 | 1.361 | 16 | 0 | 3.251 | 64 | 0 | 0.422 | 16 | 2 | 2.509 |
| | equalDist | 1 | 0 | 1.365 | 2048 | 0 | 10.854 | 1024 | 0 | 0.303 | 1024 | 0 | 2.332 |
| | equalDist2 | 1 | 0 | 1.570 | 512 | 0 | 8.050 | 1024 | 0 | 0.315 | 64 | 0 | 2.123 |
| | equalDist3 | 1 | 0 | 1.489 | 2048 | 0 | 2.764 | 1024 | 0 | 0.279 | 256 | 0 | 2.196 |
| | equalDist4 | 1024 | 0 | 2.819 | 64 | 0 | 6.029 | 512 | 0 | 0.214 | 2048 | 0 | 1.355 |
| | rand | 256 | 0 | 0.496 | 2048 | 0 | 2.248 | 2048 | 0 | 0.185 | 2048 | 0 | 0.752 |
| HRB | simple | 2048 | 0 | 0.491 | 4 | 0 | 9.667 | 256 | 0 | 0.225 | 2048 | 0 | 0.705 |
| | minVar | 1 | 0 | 1.500 | 64 | 0 | 2.319 | 256 | 0 | 0.309 | 16 | 1 | 2.801 |
| | equalDist | 1 | 0 | 1.305 | 256 | 0 | 10.635 | 1024 | 0 | 0.638 | 16 | 4 | 4.055 |
| | equalDist2 | 1 | 0 | 1.549 | 2048 | 0 | 6.790 | 16 | 0 | 0.457 | 16 | 2 | 3.445 |
| | equalDist3 | 1 | 0 | 1.405 | 1024 | 0 | 2.292 | 16 | 0 | 0.537 | 16 | 2 | 2.656 |
| | equalDist4 | 1 | 0 | 1.511 | 512 | 0 | 1.829 | 64 | 0 | 0.483 | 2048 | 0 | 2.024 |
| | rand | 2048 | 0 | 0.451 | 4 | 0 | 6.122 | 64 | 0 | 0.666 | 1024 | 1 | 2.165 |
| QB | simple | 1 | 0 | 1.469 | 16 | 0 | 10.076 | 256 | 0 | 0.297 | 256 | 1 | 3.164 |
| | minVar | 2048 | 0 | 0.050 | 1024 | 0 | 1.566 | 64 | 0 | 0.210 | 64 | 1 | 1.062 |
| | equalDist | 4 | 0 | 1.174 | 2048 | 0 | 8.134 | 2048 | 0 | 0.144 | 2048 | 0 | 0.583 |
| | equalDist2 | 2048 | 0 | 0.736 | 2048 | 0 | 4.405 | 1024 | 0 | 0.145 | 2048 | 0 | 0.539 |
| | **equalDist3** | **2048** | **0** | **0.042** | **2048** | **0** | **1.771** | **512** | **0** | **0.148** | **2048** | **0** | **0.412** |
| | **equalDist4** | **2048** | **0** | **0.130** | **512** | **0** | **1.754** | **512** | **0** | **0.134** | **512** | **0** | **0.437** |
| | rand | 1 | 0 | 1.295 | 256 | 0 | 6.048 | 16 | 0 | 0.740 | 16 | 2 | 5.988 |
| CTX | rand | 4 | 0 | 1.381 | 4 | 0 | 5.030 | 16 | 0 | 0.540 | 1024 | 1 | 2.442 |
| | rel | 1 | 0 | 1.472 | 64 | 0 | 4.021 | 64 | 0 | 0.424 | 64 | 6 | 4.349 |
| **RAND** | **rand** | **2048** | **0** | **0.104** | **1024** | **0** | **1.501** | **1024** | **0** | **0.143** | **1024** | **0** | **0.513** |

**(a)**

**Table (b): iB-10, t-1200sec, LARGE**

| Class | Scheme | DBN nAbs | Fail | Avg. Error | Grids nAbs | Fail | Avg. Error | Linkage-Type4 nAbs | Fail | Avg. Error | Promedas nAbs | Fail | Avg. Error |
|---|---|---|---|---|---|---|---|---|---|---|---|---|---|
| QB | simple | 1 | 0 | 6.540 | 16 | 0 | 197.931 | 2048 | 13 | 48.681 | 4 | 34 | 11.919 |
| | minVar | 2048 | 0 | 1.837 | 1024 | 0 | 28.423 | 256 | 31 | 93.058 | 16 | 13 | 5.403 |
| | equalDist | 512 | 0 | 5.423 | 2048 | 0 | 118.547 | 2048 | 22 | 46.196 | 512 | 15 | 5.960 |
| | equalDist2 | 2048 | 0 | 3.813 | 2048 | 0 | 91.994 | 1024 | 21 | 40.310 | 2048 | 12 | 4.982 |
| | **equalDist3** | **2048** | **0** | **1.645** | **2048** | **0** | **19.277** | **1024** | **20** | **37.490** | **256** | **5** | **2.560** |
| | **equalDist4** | **2048** | **0** | **1.643** | **2048** | **0** | **18.866** | **2048** | **16** | **30.512** | **512** | **5** | **2.476** |
| | rand | 4 | 0 | 6.292 | 16 | 0 | 163.973 | 256 | 17 | 156.992 | 4 | 28 | 11.532 |
| CTX | rand | 64 | 0 | 5.710 | 512 | 0 | 111.104 | 2048 | 53 | 194.741 | 256 | 0 | 3.222 |
| | rel | 1 | 0 | 6.267 | 1024 | 0 | 80.633 | 1024 | 37 | 129.189 | 16 | 34 | 11.247 |
| **RAND** | **rand** | **2048** | **0** | **2.123** | **2048** | **0** | **19.053** | **1024** | **19** | **33.804** | **1024** | **10** | **3.936** |

**(b)**

**Table 3: Summary Comparison.** Each table shows the Abstraction Class (*Class*), Partitioning Scheme (*Scheme*), bound on the number of abstract states per level (*nAbs*), number of problems for which a positive solution could not be estimated (*Fail*), and average $\log_{10} Z$ error (*Avg. Error*) across Exact problems (subtable (a)) and LARGE problems (subtable (b)) in each benchmark. Color bars visualize error magnitudes. We hightliht the best performing algorithms: those for which: (1) difference in total average error (summed across the benchmarks) with respect to the best such total was less than 15% of the best, and (2) within each individual benchmark, the difference in average error with respect to the best average error was less than 35% of the best. (An exception to the latter criterion was granted to Exact DBN, on which the best average error from equalDistQB3 was unusually low).

**Table 4: iB-5, m-100, Exact**

| Class | Scheme | nAbs | DBN Fail | Avg. Error | Grids Fail | Avg. Error | Pedigree Fail | Avg. Error | Promedas Fail | Avg. Error |
|---|---|---|---|---|---|---|---|---|---|---|
| QB | simpleQB | 256 | 0 | 5.350 | 0 | 17.406 | 0 | 1.059 | 14 | 9.659 |
| | **minVarQB** | **256** | **0** | **0.111** | **0** | **1.911** | **0** | **0.223** | **1** | **1.634** |
| | equalDist | 256 | 0 | 5.619 | 0 | 15.533 | 1 | 0.858 | 13 | 5.420 |
| | equalDist2 | 256 | 0 | 2.319 | 0 | 11.220 | 0 | 0.563 | 6 | 3.479 |
| | **equalDist3** | **256** | **0** | **0.173** | **0** | **3.615** | **0** | **0.206** | **1** | **1.473** |
| | **equalDist4** | **256** | **0** | **0.277** | **0** | **2.305** | **0** | **0.180** | **1** | **1.373** |
| | randQB | 256 | 0 | 4.982 | 0 | 12.653 | 0 | 3.211 | 13 | 19.441 |
| CTX | rand | 256 | 0 | 3.587 | 0 | 9.568 | 2 | 4.695 | 3 | 14.386 |
| | rel | 256 | 0 | 5.265 | 0 | 8.013 | 36 | 1.097 | 36 | 10.845 |
| **RAND** | **rand** | **256** | **0** | **0.288** | **0** | **2.464** | **0** | **0.325** | **3** | **2.570** |

**Table 4: 100-Sample Comparison.** For abstraction granularity of $nAbs = 256$, aggregated statistics (as described in Table 3) for Exact problems of each benchmark with each algorithm allotted 100 samples.

SampleSearch (IJGP-ss) [Gogate and Dechter, 2011], and Dynamic Importance Sampling [Lou et al., 2019]. Thus, superior performance against CTX schemes implicitly indicates competitiveness against the these other methods.

**Abstraction Quality of the QB Schemes.** When drawing an equal number of samples with the same abstraction gran-

**Table 5: iB-5, t-300sec, Exact**

| Class | Scheme | nAbs | DBN Fail | Avg. Error | Grids Fail | Avg. Error | Pedigree Fail | Avg. Error | Promedas Fail | Avg. Error |
|---|---|---|---|---|---|---|---|---|---|---|
| QB | minVar | 4 | 0 | 1.684 | 0 | 3.622 | 0 | 1.434 | 2 | 2.518 |
| | | 64 | 0 | 0.180 | 0 | 1.897 | 0 | 0.210 | 1 | 1.062 |
| | | 1024 | 0 | 0.060 | 0 | 1.566 | 0 | 0.479 | 2 | 1.837 |
| | equalDist3 | 4 | 0 | 1.594 | 0 | 5.861 | 0 | 1.668 | 1 | 1.804 |
| | | 64 | 0 | 0.236 | 0 | 2.570 | 0 | 0.221 | 0 | 0.570 |
| | | 1024 | 0 | 0.051 | 0 | 1.844 | 0 | 0.155 | 0 | 0.462 |
| | equalDist4 | 4 | 0 | 1.371 | 0 | 5.988 | 0 | 1.648 | 1 | 1.678 |
| | | 64 | 0 | 0.215 | 0 | 2.438 | 0 | 0.231 | 0 | 0.596 |
| | | 1024 | 0 | 0.150 | 0 | 1.891 | 0 | 0.150 | 0 | 0.455 |
| CTX | rand | 4 | 0 | 1.381 | 0 | 5.030 | 0 | 1.852 | 7 | 4.643 |
| | | 64 | 0 | 1.763 | 0 | 5.950 | 0 | 0.598 | 1 | 2.659 |
| | | 1024 | 0 | 2.007 | 0 | 5.513 | 0 | 1.114 | 1 | 2.442 |
| | rel | 4 | 0 | 1.850 | 0 | 5.933 | 0 | 1.332 | 10 | 5.729 |
| | | 64 | 0 | 3.510 | 0 | 4.021 | 0 | 0.424 | 6 | 4.349 |
| | | 1024 | 0 | 5.086 | 0 | 5.136 | 0 | 1.041 | 15 | 6.688 |
| RAND | rand | 4 | 0 | 1.018 | 0 | 4.329 | 0 | 1.705 | 2 | 2.947 |
| | | 64 | 0 | 0.418 | 0 | 2.094 | 0 | 0.212 | 0 | 0.757 |
| | | 1024 | 0 | 0.120 | 0 | 1.501 | 0 | 0.143 | 0 | 0.513 |

**Table 5: Varying nAbs.** Average error when using $nAbs \in \{4, 64, 1024\}$ for minVarQB, equalDistQB3, equalDistQB4, the CTX based algorithms, and RAND, each with iB-5 and time limit of 300 sec.

ularity of $nAbs = 256$ (Table 4), QB with *equalDist3* and *equalDist4* and RAND are well performing as seen when

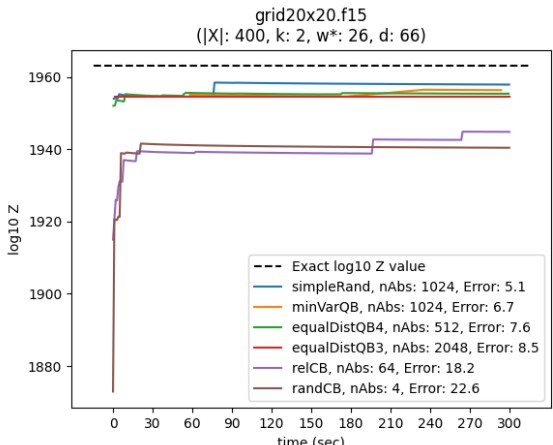

**Figure 6:** Z estimates from various algorithms versus time on Exact Grids problem grid20x20.f15 using $iB = 5$. The dashed black line shows the true Z value.

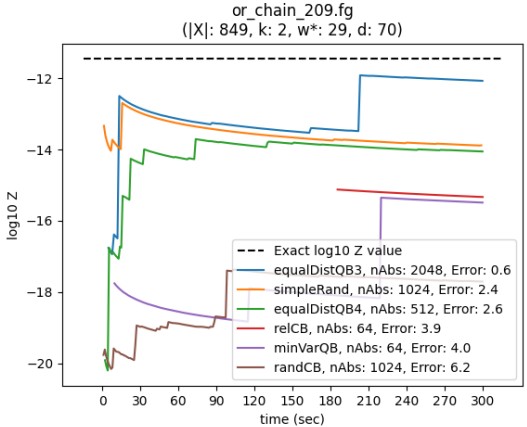

**Figure 7:** Z estimates from various algorithms versus time on Exact Promedas problem or_chain_209.fg using $iB = 5$. The dashed black line shows the true Z value.

using a time limit (Table 3). A key difference is that QB with *minVar*, which had showed slightly worse performance under a time limit, is now best. This in part explains the success of QB *equalDist3* and *equalDist4*, which try to emulate QB *minVar* while using faster greedy strategies.

**Anytime Behavior.** Figure 6 and Figure 7 show that Abstraction Sampling estimates continue to improve as time progresses. We also notice that estimates are often underestimates that increase over time, a common phenomenon of importance sampling due to the proposal distribution's tails.

**Choice of Abstraction Granularity.** From Table 5 we see that for the well performing QB *equalDist3* and *equalDist4* schemes and for the RAND scheme there is a trend that greater $nAbs$ improves performance. Figure 5 further supports this for QB with *equalDist4*, for which plots of QB *equalDist3* and RAND have similar profiles (omitted for brevity). However in Figure 4 and Table 5 we see that for *minVar* error begins to increase when $nAbs$ becomes too high. This can be explained by the higher computational cost of forming *minVar* abstractions (which is more time

| | HB | HRB | QB |
|---|---|---|---|
| **simple** | 2.75 | 1.12 | 0.72 |
| **minVar** | 1.05 | 1.13 | 2.95 |
| **equalDist** | 0.75 | 0.59 | 1.16 |
| **equalDist2** | 0.84 | 0.75 | 1.82 |
| **equalDist3** | 1.20 | 1.01 | 4.05 |
| **equalDist4** | 0.87 | 1.14 | 3.90 |
| **rand** | 2.41 | 0.93 | 0.60 |

**Figure 8: Performance Matrix**. Relative average performance of value-based schemes vs. existing context-based abstractions. Values $> 1.00$ indicate superior performance.

consuming), leaving less time for probe generation.

**Summary of Results.** Experiments show the QB scheme with *equalDist3* or *equalDistQB4* and RAND performing the best of the newly proposed abstraction functions, significantly outperforming the former state-of-the-art (Figure 8). These schemes tend to improve as the abstraction granularity $nAbs$ increases up to a point, past which we see little difference in performance. Thus, our study suggests that these three abstraction schemes should be the first choice when using AOAS, and be used with the largest $nAbs$ feasible.

# 7 CONCLUSION

This exploration of abstraction functions for use with AND/OR Abstraction Sampling (AS) featured a new value-based abstraction framework, introducing three abstraction classes: HB, QB, and HRB each defined by real-valued functions that aim to capture informative elements from search and sampling to guide abstractions and improve AS performance. Each class was tested with each of seven node partitioning schemes to form twenty-one new abstraction functions. Additionally, a new purely randomized abstraction scheme, RAND, was presented that places nodes into equal cardinality abstract states completely at random.

Results from an extensive empirical evaluation on over 400 benchmark problems show two of the QB based schemes (*equalDistQB3*, and *equalDistQB4*) and the RAND scheme having superior performance consistently and throughout all benchmarks. In particular, performance was significantly improved relative to former state-of-the-art context-based abstractions, and thus also implicitly against Importance Sampling, Weighted Mini-Bucket Importance Sampling, IJGP-SampleSearch, and Dynamic Importance Sampling.

Based on this study and earlier findings, we believe that AOAS is one of the best schemes for estimating the partition function to date. Future work will explore adjusting the abstraction schemes to problem instances through learning and also the potential for applying adaptive sampling.

### Acknowledgements

Thank you to the reviewers for their valuable comments and suggestions. This work was supported in part by NSF grants IIS-2008516 and CNS-2321786.

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

# Value-Based Abstraction Functions for Abstraction Sampling
## (Supplemental Materials)

**Bobak Pezeshki**[1]      **Kalev Kask**[1]      **Alexander Ihler**[1]      **Rina Dechter**[1]

[1]University of California, Irvine

## Abstract

For revised supplemental materials, please visit `https://ics.uci.edu/~dechter/publications.html`. This document includes supplemental background, descriptions, details, and results in extension to the main paper. Given its size, we suggest using the table of contents to navigate. For an additional background on graphical models, AND/OR search trees, and variable elimination, please view the EXTENDED BACKGROUND supplemental document.

## CONTENTS

# 1 AOAS BACKGROUND

Taken with permission directly from Kask et al. [2020].

## 1.1 SAMPLE ALGORITHM TRACE

Here we show a trace of abstraction sampling using the AOAS algorithm using an abstraction function that groups AND nodes of the same domain value together in an abstract state.

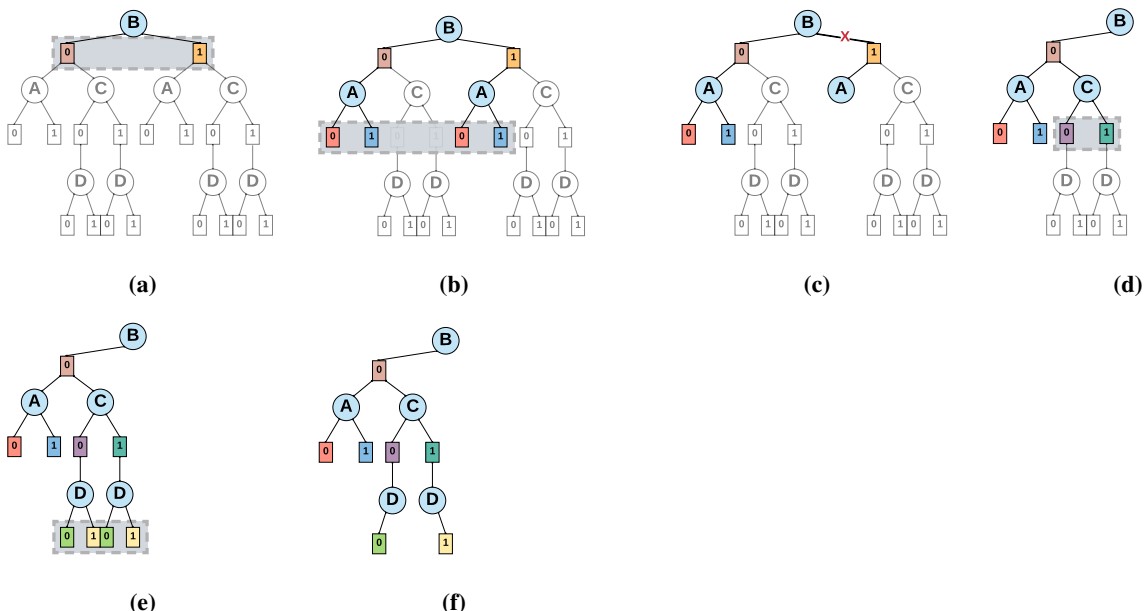

**Figure 1:** From Kask et al. [2020], a sample trace of AOAS following DFS ordering $B \to A \to C \to D$. Transparent nodes indicate portions of the reachable search space yet to be explored. Gray boxes indicate nodes considered for abstraction. Nodes with the same domain values (also indicated by the same color) are abstracted into the same abstract state. Only one node of each color is stochastically selected as a representative for its respective abstract state. Step (c) shows an optional optional pruning step. Step (f) shows the final example probe capturing four full configurations: $B = 0, A = 0, C = 0, D = 0, B = 0, A = 1, C = 0, D = 0, B = 0, A = 0, C = 1, D = 1, B = 0, A = 1, C = 1, D = 1$.

Starting with variable $B$ (Figure 1a), each node belongs to a different abstraction and is therefore kept. Next, we expand to $A$ and abstract across its nodes (Figure 1b). Not restricted to *proper* abstractions, we partition across *all* nodes of $A$, regardless of whether they fall under $B=0$ or $B=1$. We see two nodes in each abstract state (denoted by the red and blue coloring). Next we calculate their respective proposals (line 21). Note that the proposal of each node $n$ relies on $r(n)$ (line 15), which captures the values of the nodes in its $Out(path(n))$, in this case nodes of $C$. Since the nodes of $C$ have not been expanded yet, we use their heuristic values as an approximation of their values. We then stochastically choose a representative from each abstract state (line 23). Suppose that both red and blue representatives are stochastically chosen from under $B=0$ (Figure 1c). Since $A$ has no descendant, we backtrack to $B$, updating its node values (line 33) and performing a pruning step (line 31). In pruning, we remove AND nodes of $B$ that do not extend to AND nodes of $A$, and thus prune $B=1$ (denoted by the red "X" in Figure 1c), in order to ensure formation of proper AND/OR probes. Finally, we expand and abstract $C$ and $D$ (Figures 1d-1f). The $r(n)$ for $D$'s nodes is inherited from the $r(n_C)$ of its respective $n_C$ parent. We backtrack from $D$ to the root updating values (no further pruning was necessary). The result is a valid probe (Figure 1f) containing four solutions: $(B=0, A=0, C=0, D=0)$, $(B=0, A=0, C=1, D=1)$, $(B=0, A=1, C=0, D=0)$, and $(B=0, A=1, C=1, D=1)$. We estimate the partition function by computing $\hat{Z}(B)$.

## 1.2 DETAILED ALGORITHM

**Algorithm 1:** AOAS.

**Input:** Graphical model $\mathcal{M} = (\mathbf{X}, \mathbf{D}, \mathbf{\Phi})$, a pseudo tree $\mathcal{T}$ for $\mathcal{M}$ rooted at a dummy singleton variable $D$, an abstraction function $a$, heuristic function $h$. For any node $n$, $g(n) =$ its path cost, $w(n) =$ its importance weight, and $\hat{Z}(n) =$ its estimated value (initialized to $h(n)$).

**Output:** $\hat{Z}_{\mathcal{M}}$, an estimate of the partition function of $\mathcal{M}$

```
 1 Function AOAS (𝒯, h, a)
 2 begin
 3     PROBE ← n_D, g(n_D), w(n_D), r(n_D), Ẑ(n_D) ← 1
 4     STACK ← push(empty stack, D)
 5     while STACK is not empty do
 6         X ← top(STACK)
 7         if X has unvisited children in 𝒯 then
 8             Y ← the next unvisited child of X
 9             foreach n_X ∈ PROBE do
10                 PROBE ← PROBE expanded from n_X to Y
11                 F'_Y ← newly added AND nodes of Y ∈ PROBE
12                 foreach n_Y ∈ F'_Y do
13                     w(n_Y) ← w(n_X)
14                     g(n_Y) ← g(n_X) · c(n_Y)
15                     r(n_Y) ← r(n_X) · ∏_{S≠Y∈ch_𝒯(X)} V̂(S_{n_X})
16                 end
17             end
18             A ← {A_i | A_i = {n_Y ∈ PROBE | a(n) = i}}
19             foreach A_i ∈ A do
20                 foreach n ∈ A_i do
21                     p(n) ← (w(n)·g(n)·h(n)·r(n)) / (∑_{m∈A_i} w(m)·g(m)·h(m)·r(m))
22                 end
23                 n_{Y_i} ∝_p A_i ;                                    // randomly select
24                 w(n_{Y_i}) ← w(n_{Y_i})/p(n_{Y_i})
25                 Ẑ(n_{Y_i}) ← 1
26                 PROBE ← PROBE \ A_i ∪ {n_{Y_i}}
27             end
28             push(STACK, Y)
29         else
30             pop(STACK), W ← top(STACK)
31             PROBE ← PROBE s.t. all n_W without descendants are pruned
32             foreach n_W in PROBE do
33                 Ẑ(n_W) ← Ẑ(n_W) · ∑_{n_X←child(n_W)} Ẑ(n_X) · c(n_X) · w(n_X)/w(n_W)
34             end
35             if X = D then Ẑ_𝓜 = Ẑ(D);
36         end
37     end
38     return Ẑ_𝓜
39 end
```

## 2 PROBE SIZE VARIABILITY

Even with the same abstraction function and granularity (ie. allowed number of abstract states per level), probe sizes can vary greatly. One reason for this is due to abstractions causing nodes from certain branches of the probe to replaced by representative from other branch, and thus the current branch will no longer be extended. We provide a paired example in Figure 3 and Figure 4 where in both cases the probes are constructed according to the pseudo tree shown in Figure 2, an abstraction function is used that groups nodes with the same domain value together (indicated by yellow coloring for grouping of nodes with a domain value of 0 and blue coloring grouping nodes together that have domain value of 1) is used, and the abstraction granularity is set to $nAbs = 2$ (meaning that nodes are abstracted into at most two abstract states).

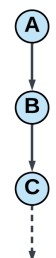

**Figure 2:** A linear psuedo tree.

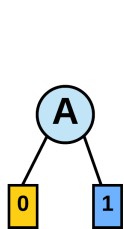

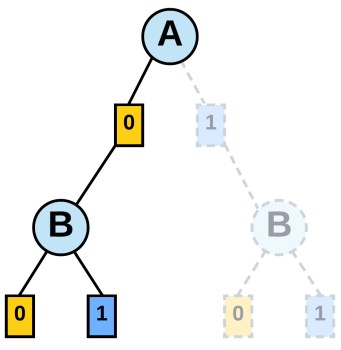

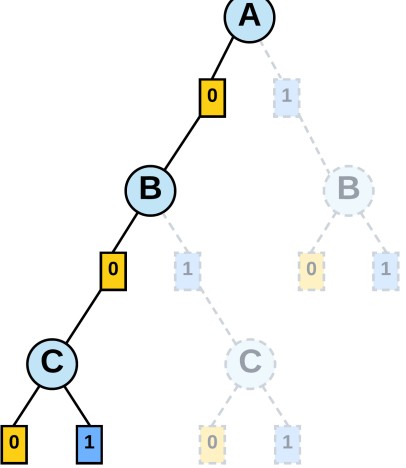

**(a)** Variable $A$ is expanded. Each node is placed into a separate abstract state and each is selected to represent their respective abstract state.

**(b)** Variable $B$ is expanded from each existing node of $A$. $B$ nodes with domain value 0 are joined together into an abstract state (yellow); $B$ nodes with domain value 1 constitute a different abstract state (blue). For each resulting abstract state, the corresponding node underneath the branch of $A \leftarrow 0$ is stochastically selected as the representative. As there are no selected representatives underneath the branch of $A \leftarrow 1$, those nodes will no longer be extended (and can be pruned).

**(c)** Variable $C$ is expanded from each representative node of $B$. $C$ nodes with domain value 0 are joined together into an abstract state (yellow); $C$ nodes with domain value 1 constitute a different abstract state (blue). For each resulting abstract state, the corresponding node underneath the branch of $A \leftarrow 0, B \leftarrow 0$ is stochastically selected as the representative. As there are no selected representatives underneath the branch of $A \leftarrow 0, B \leftarrow 1$, those nodes will no longer be extended (and can be pruned).

**Figure 3:** An example of a "skewed" probe construction following the pseudo tree in Figure 2, using an abstraction function that groups nodes of the same domain value into the same abstract state, and using a granularity of $nAbs = 2$. At each level, representatives of all abstract states are chosen under the same single branch, thus only extending only one path in the probe.

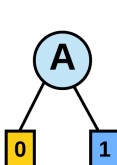

**(a)** Variable $A$ is expanded. Each node is placed into a separate abstract state and each is selected to represent their respective abstract state.

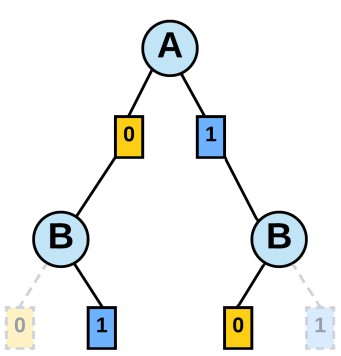

**(b)** Variable $B$ is expanded from each existing node of $A$. $B$ nodes with domain value 0 are joined together into an abstract state (yellow); $B$ nodes with domain value 1 constitute a different abstract state (blue). The stochastically selected representative from the $B = 1$ abstract state ends up under the $A \leftarrow 0$ branch while the representative from the $B = 0$ abstract state is selected from under $A \leftarrow 1$. As a result, both $A \leftarrow 0$ and $A \leftarrow 1$ branches have an extension to a node from $B$ and will continue to be extended.

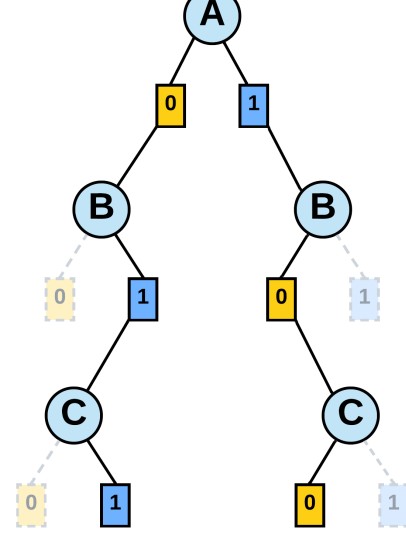

**(c)** Variable $C$ is expanded from each existing node of $B$. $C$ nodes with domain value 0 are joined together into an abstract state (yellow); $C$ nodes with domain value 1 constitute a different abstract state (blue). The stochastically selected representative from the $C = 1$ abstract state ends up under the $A \leftarrow 0, B \leftarrow 1$ branch while the representative from the $C = 0$ abstract state is selected from under $A \leftarrow 1, B \leftarrow 0$. As a result, both $A \leftarrow 0, B \leftarrow 1$ and $A \leftarrow 1, B \leftarrow 0$ branches have an extension to a node from $C$ and will continue to be extended.

**Figure 4:** An example of a "balanced" probe construction following the pseudo tree in Figure 2, using an abstraction function that groups nodes of the same domain value into the same abstract state, and using a granularity of $nAbs = 2$. At each level, representatives of all abstract states are chosen under the same single branch, thus only extending only one path in the probe.

# 3 EXACT ABSTRACTION PROOFS

**Required Definitions.**

**Definition 3.0.0.1** (Abstraction Function $h(n)$ vs. $Z(n)$ Proportionality)
*An abstraction function $a(n)$ maintains $h(n)$ vs. $Z(n)$ proportionality if, for every abstract state $A_i$ formed by $a(n)$, $\forall n \in A_i, h(n) = \alpha \, Z(n)$, for some constant $\alpha$ specific to $A_i$.*

**Definition 3.0.0.2** (Abstraction Function $h(n)r(n)$ vs. $Z(n)R(n)$ Proportionality)
*An abstraction function $a(n)$ maintains $h(n)r(n)$ vs. $Z(n)R(n)$ proportionality if, for every abstract state $A_i$ formed by $a(n)$, $\forall n \in A_i, h(n)r(n) = \alpha \, Z(n)R(n)$, for some constant $\alpha$ specific to $A_i$.*

**Definition 3.0.0.3** (Exact Abstraction Function)
*An abstraction function $a(.)$ is exact for an abstraction sampling algorithm, AS, if use of $a(.)$ with AS always leads to AS estimates having zero variance and $\hat{Z} = Z$ for every AS probe.*

## 3.1 ORAS

**Theorem 3.1.0.1** (ORAS Exact Abstractions from $h(n)$ vs. $Z(n)$ Proportionality)
*If an abstraction function $a(.)$ maintains $h(n)$ vs. $Z(n)$ Proportionality, then it is an exact abstraction function for ORAS.*

*Proof.* We know that if we were to use exhaustive search, we would arrive at the true $Z$ value. We use a proof by induction that assumes that after each abstraction step we will compute the rest of the probe exactly using exhaustive search. Thus, if abstractions are performed layer by layer down from the root, after each abstraction we know that $Z(n')$ will be computed exactly for the selected node $n'$.

We denote the estimate that would be generated by a probe constructed after $t$ time steps as $\hat{Z}^{(t)}(PROBE)$. (As we will describe, each time step will correspond to an abstraction step). As a base case, $\hat{Z}^{(t=0)}(PROBE) = Z$ since all values will be computed exactly via exhaustive search. In the inductive step, we will show that after each time step $t$, if instead of using exhaustive search immediately, we first perform an abstraction on the current level of the probe, the resulting estimate of the newly abstracted probe $\hat{Z}^{(t+1)}(PROBE)$ will remain unchanged. Namely, we will show that

$$\hat{Z}^{(t)}(PROBE) - \hat{Z}^{(t+1)}(PROBE) = 0$$

This shows that the abstractions maintain exactness of the probe's estimate.

Starting from the left hand side

$$LHS = \hat{Z}^{(t)}(PROBE) - \hat{Z}^{(t+1)}(PROBE)$$

We note the difference in the overall probe estimates during an Abstraction Sampling is due to the change in the probe estimate that results from each individual abstraction step (namely selection and reweighing of a representative node $n'$ from an abstract state $A_i$). Thus for our time steps, we will focus on the difference in value resulting from a single arbitrary abstraction step.

$$= \sum_{n \in A_i} w^{(t)}(n)g(n)Z(n) - w^{(t+1)}(n')g(n')Z(n')$$

Above, the left term shows the contribution to the partition function due to nodes of abstract state $A_i$ (still assuming we will perform exhaustive search below each one), and the right term is the contribution of a selected node $n'$ after abstraction (note the adjustment to the selected node's weight).

Using the fact that $w^{(t+1)}(n') = \frac{w^{(t)}(n')}{p(n')}$ (from the importance weight modification), we now get

$$= \sum_{n \in A_i} w^{(t)}(n)g(n)Z(n) - \frac{w^{(t)}(n')}{p(n')}g(n')Z(n')$$

(Note that $p(n')$ cannot be zero, otherwise $n'$ would not have been selected).

Noting that for $p(n') = \frac{w^{(t)}(n')g(n')h(n')}{\sum_{n \in A_i} w^{(t)}(n)g(n)h(n)}$ and substituting we get

$$= \sum_{n \in A_i} w^{(t)}(n)g(n)Z(n)$$

$$- w^{(t)}(n')g(n')Z(n')\frac{\sum_{n \in A_i} w^{(t)}(n)g(n)h(n)}{w^{(t)}(n')g(n')h(n')}$$

$$= \sum_{n \in A_i} w^{(t)}(n)g(n)Z(n) - \frac{Z(n')}{h(n')} \sum_{n \in A_i} w^{(t)}(n)g(n)h(n)$$

Now, per our assumption, $\forall n \in A_i$, let $h(n) = \alpha\, Z(n)$, where $\alpha$ is the proportionality constant by which $h(n)$ differs from $Z(n)$. Then

$$= \sum_{n \in A_i} w^{(t)}(n)g(n)Z(n) - \frac{Z(n')}{\alpha\, Z(n')} \sum_{n \in A_i} w^{(t)}(n)g(n)\, \alpha\, Z(n)$$

$$= \sum_{n \in A_i} w^{(t)}(n)g(n)Z(n) - \frac{\alpha}{\alpha} \sum_{n \in A_i} w^{(t)}(n)g(n)Z(n)$$

$$= \sum_{n \in A_i} w^{(t)}(n)g(n)Z(n) - \sum_{n \in A_i} w^{(t)}(n)g(n)Z(n)$$

$$= 0 = RHS$$

$\square$ $\square$

## 3.2 AOAS

**Theorem 3.2.0.1** (AOAS Exact Abstractions from $h(n)r(n)$ vs. $Z(n)R(n)$ Proportionality)
*If an abstraction function $a(.)$ maintains $h(n)r(n)$ vs. $Z(n)R(n)$ Proportionality, then it is an exact abstraction function for AOAS.*

*Proof.* We know that if we were to use exhaustive search, we would arrive at the true $Z$ value. We use a proof by induction that assumes that after each abstraction step we will compute the rest of the probe exactly using exhaustive search. Thus, if abstractions are performed layer by layer down from the root, after each abstraction we know that $Z(n')$ will be computed exactly for the selected node $n'$. We also assume that, $R(n)$ for every node will be computed exactly. This assumption holds true before we perform any abstractions (as everything is computed exactly via exhaustive search) and continues to hold if we can show that, after each abstraction step, the resulting estimates remains unchanged (and thus remains exact).

We denote the estimate that would be generated by a probe constructed after $t$ time steps as $\hat{Z}^{(t)}(PROBE)$. (As we will describe, each time step will correspond to an abstraction step). As a base case, $\hat{Z}^{(t=0)}(PROBE) = Z$ since all values will be computed exactly via exhaustive search. In the inductive step, we will show that after each time step $t$, if instead of using exhaustive search immediately, we first perform an abstraction on the current level of the probe, the resulting estimate of the newly abstracted probe $\hat{Z}^{(t+1)}(PROBE)$ will remain unchanged. Namely, we will show that
$$\hat{Z}^{(t)}(PROBE) - \hat{Z}^{(t+1)}(PROBE) = 0$$
This shows that the abstractions maintain exactness of the probe's estimate.

Starting from the left hand side
$$LHS = \hat{Z}^{(t)}(PROBE) - \hat{Z}^{(t+1)}(PROBE)$$

We note the difference in the overall probe estimates during an Abstraction Sampling is due to the change in the probe estimate that results from each individual abstraction step (namely due to the selection and reweighing of a representative node $n'$ from an abstract state $A_i$). Thus for our time steps, we will focus on the difference in value resulting from a single arbitrary abstraction step.
$$= \sum_{n \in A_i} w^{(t)}(n)g(n)Z(n)R(n) - w^{(t+1)}(n')g(n')Z(n')R(n')$$

Above, the left term shows the contribution to the partition function due to nodes of abstract state $A_i$ (still assuming we will perform exhaustive search below each one), and the right term is the contribution of a selected node $n'$ after abstraction (note the adjustment to the selected node's weight).

Using the fact that $w^{(t+1)}(n') = \frac{w^{(t)}(n')}{p(n')}$ (from the importance weight modification), we now get

$$= \sum_{n \in A_i} w^{(t)}(n)g(n)Z(n)R(n) - \frac{w^{(t)}(n')}{p(n')}g(n')Z(n')R(n')$$

(Note that $p(n')$ cannot be zero, otherwise $n'$ would not have been selected).

Noting that for $p(n') = \frac{w^{(t)}(n')g(n')h(n')r(n')}{\sum_{n \in A_i} w^{(t)}(n)g(n)h(n)r(n')}$ and substituting we get

$$= \sum_{n \in A_i} w^{(t)}(n)g(n)Z(n)R(n)$$

$$- w^{(t)}(n')g(n')Z(n')R(n')\frac{\sum_{n \in A_i} w^{(t)}(n)g(n)h(n)r(n)}{w^{(t)}(n')g(n')h(n')r(n')}$$

$$= \sum_{n \in A_i} w^{(t)}(n)g(n)Z(n)R(n)$$

$$- \frac{Z(n')R(n')}{h(n')r(n')} \sum_{n \in A_i} w^{(t)}(n)g(n)h(n)r(n)$$

Now, per our assumption, $\forall n \in A_i$, let $h(n)r(n) = \alpha\, Z(n)R(n)$, where $\alpha$ is the proportionality constant by which $h(n)r(n)$ differs from $Z(n)R(n)$. Then

$$= \sum_{n \in A_i} w^{(t)}(n)g(n)Z(n)R(n)$$

$$- \frac{Z(n')R(n')}{\alpha\, Z(n')R(n')} \sum_{n \in A_i} w^{(t)}(n)g(n)\, \alpha\, Z(n)R(n)$$

$$= \sum_{n \in A_i} w^{(t)}(n)g(n)Z(n)R(n)$$

$$- \frac{\alpha}{\alpha} \sum_{n \in A_i} w^{(t)}(n)g(n)Z(n)R(n)$$

$$= \sum_{n \in A_i} w^{(t)}(n)g(n)Z(n)R(n) - \sum_{n \in A_i} w^{(t)}(n)g(n)Z(n)R(n)$$

$$= 0 = RHS$$

$$\square \qquad\qquad\qquad\qquad\qquad\qquad\qquad \square$$

# 4 PARADIGMS INTUITING ABSTRACTION STRATEGIES

Next we review concepts from search and sampling that offer paradigms from which we draw ideas for abstraction functions.

## 4.1 SEARCH PARADIGMS

In [tree] search, one can merge nodes that have the same value to produce a more efficient graph search Mateescu et al. [2008]. Abstraction functions by Broka et al. [2018] focused on this paradigm and approached it by using the concept of a node's context - the assignments to the smallest subset of a node's ancestor variables that dictates its value. Due to the potentially large context size for variables, and consequently the exponentially high number of combinations of assignments to the context, the full context of variables could not be used in most cases. Broka et al. [2018] resolved this by creating two context-based abstraction functions that were relaxed to allow nodes with different contexts to be grouped in the same abstract state. However, sharing the same partial context does not necessarily imply the same, nor even similar, node values. Our new Heuristic-Based abstractions hope to provide more accurate abstractions based on the same ideology.

## 4.2 SAMPLING PARADIGMS

Consider wanting to compute the $\mathbb{E}_{p^*}[f(x)] = \sum_x f(x)p^*(x)$ for a distribution $p^*(.)$ over a variable $X$ that is difficult to sample from but easy to evaluate, and given a positive value function $f(x)$. Using a proposal distribution $p(.)$ that is easy to sample from, and noticing the equivalency of the target quantity with $\sum_x \frac{f(x)p^*(x)}{p(x)}p(x)$, we can be estimate the quantity by importance sampling by drawing $m$ samples to estimate the equivalent quantity $\mathbb{E}_p[f(x)\frac{p^*(x)}{p(x)}] \approx \frac{1}{m}\sum_{j=1}^m f(x^{(j)})\frac{p^*(x^{(j)})}{p(x^{(j)})}, x^{(j)}\overset{\text{iid}}{\sim}p$. it is well known that importance sampling achieves zero variance when 1) $p(x) = 0 \implies p^*(x) = 0$, and 2) otherwise $p(x)$ is proportional to $p^*(x)f(x)$ Kahn and Marshall [1953], Owen [2013].

**Lemma 4.2.0.1** (Importance Sampling Exact Proposal Based on Proportionality with Target Distribution)
*Given a distribution $p^*(.)$ over a variable $X$ that is easy to evaluate, and given a positive value function $f(x)$, importance sampling to estimate $\mathbb{E}_{p^*}[f(x)]$ achieves zero variance when using a proposal function $p(.)$ such that 1) $p(n) = 0 \implies p^*(n)f(n) = 0$, and 2) $p(n) \propto p^*(n)f(n)$, otherwise.*

Note that we can also use importance sampling to simply compute $\sum_x f(x) = \sum_x \frac{f(x)}{p(x)}p(x) = \mathbb{E}_p[\frac{f(x)}{p(x)}] \approx \frac{1}{m}\sum_{j=1}^m \frac{f(n^{(j)})}{p(x^{(j)})}, x^{(j)}\overset{\text{iid}}{\sim}p$. Note that the partition function over a graphical model, $Z = \sum_{\boldsymbol{x}} \boldsymbol{F}(\boldsymbol{x}), \boldsymbol{F}(\boldsymbol{x}) = \prod_{f\in\boldsymbol{F}} f(x)$, has the form of this task.

In fact, expanding an AND/OR search tree level-by-level, the partition function $Z$ with respect to the nodes $n$ at any variable $X$ can be written as $Z = \sum_n g(n)Z(n)R(n)$. Thus, using a proposal $p(.)$ to perform importance sampling at any level we could instead estimate

$$Z = \sum_n g(n)Z(n)R(n) = \sum_x \frac{g(n)Z(n)R(n)}{p(n)}p(n) \tag{1}$$

$$\approx \frac{1}{m}\sum_{j=1}^m \frac{g(n^{(j)})Z(n^{(j)})R(n^{(j)})}{p(n^{(j)})}, n^{(j)}\overset{\text{iid}}{\sim}p \tag{2}$$

Thus, sampling at any level would also allow for zero variance / exact computation if similarly $p(n) \propto g(n)Z(n)R(n)$.

Note that in Abstraction Sampling each abstract state involves a node selection procedure analogous to importance sampling and that AOAS uses a proposal $p(n) \propto g(n)h(n)r(n)$. $g(n)$ can always be evaluated exactly. Then assuming that $h(n) = 0 \implies Z(n) = 0$ and $r(n) = 0 \implies R(n) = 0$, it naturally follows that designing each abstract states $\boldsymbol{A_i}$ such that $\forall n \in \boldsymbol{A_i}, h(n)r(n) = \alpha\, g(n)Z(n)R(n)$, for some constant $\alpha$, we similarly achieve zero variance.

**Definition 4.2.0.1** (Abstraction Function $h(n)r(n)$ vs. $Z(n)R(n)$ Proportionality)
*An abstraction function $a(n)$ maintains $h(n)r(n)$ vs. $Z(n)R(n)$ proportionality if, for every abstract state $A_i$ formed by $a(n)$, $\forall n \in A_i, h(n)r(n) = \alpha\, Z(n)R(n)$, for some constant $\alpha$ specific to $A_i$.*

**Definition 4.2.0.2** (Exact Abstraction Function)
*An abstraction function $a(.)$ is exact for an abstraction sampling algorithm, AS, if use of $a(.)$ with AS always leads to AS estimates having zero variance and $\hat{Z} = Z$ for every AS probe.*

Thus, we can say:

**Theorem 4.2.0.2** (AOAS Exact Abstractions from $h(n)r(n)$ vs. $Z(n)R(n)$ Proportionality)
*If an abstraction function $a(.)$ maintains $h(n)r(n)$ vs. $Z(n)R(n)$ Proportionality, then it is an exact abstraction function for AOAS. (Proof in Supplemental Materials)*

Normally we neither have access to the proportionality constant $\alpha$ or even know whether nodes have the same $\alpha$. However one idea is to use the magnitude of $h(n)r(n)$ itself as a heuristic for similarities in $\alpha$. This drives the intuition for a new HR-Based class of abstractions.

Also from a sampling perspective, Rizzo [2007] showed the following about stratified importance sampling when sampling from equal area strata under the proposal:

**Proposition 4.2.0.3** (Stratified Importance Sampling Variance Reduction)
*Suppose that $M = mk$ is the number of replicates for an importance sampling estimator $\hat{\theta}^I$, and $\theta^{\hat{S}I}$ is a stratified importance sampling estimator, with estimates $\hat{\theta}_j$ for $\theta_j$ on the individual strata, each with $m$ replicates. If $Var(\hat{\theta}^I) = \sigma^2/M$ and $Var(\hat{\theta}_j) = \sigma_j^2/m$, $j = 1, ..., k$, then*

$$\sigma^2 - k \sum_{j=1}^{k} \sigma_j^2 \geq 0, \tag{3}$$

*with equality if and only if $\theta_1 = ... = \theta_k$. Hence stratification never increases variance, and there exists a stratification that reduces the variance except when [the proposal function] $g(x)$ is constant.*

Two takeaways from this proposition are that 1) we can achieve variance reduction with respect to importance sampling (analogous to Abstraction Sampling with all nodes placed into a single abstract state) by stratifying into equal area strata under the proposal, and 2) reducing the variance of each strata $\sigma_j^2$ leads to greater variance reduction. These will help drive the intuition for a new Q-Based abstraction class, as well as motivate several new partitioning schemes.

# 5 ADDITIONAL INFORMATION ABOUT VALUE-BASED ABSTRACTIONS

As described in the main paper, value-based abstraction functions consist of two parts: (1) a value function $\mu : n \to \mathbb{R}$ that assigns a real value on a positive scale to nodes $n$ that are to be abstracted, and (2) a partitioning scheme that then abstracts nodes based on $\mu(n)$. And because $\mu(n)$ are values on a positive scale (implying semantics between smaller vs. larger values), the partitioning schemes can be designed to partition the nodes in a way that maintains an ordering of $\mu(n)$. This results in what we call value-based ordered abstractions.

---

**Algorithm 2:** General Value-0Ordered Abstraction Function Scheme

---

**input** : A set of nodes $\boldsymbol{n}$ to be partitioned into abstract states; an abstraction value function $\mu(\cdot)$; a parameter $nAbs$ bounding the number of abstract states; a partitioning function $\Psi_o(\cdot)$ that partitions $\boldsymbol{n}$ into abstract states such that nodes are ordered by $\mu(n)$ according to sort-order $o$

**output** : Nodes $\boldsymbol{n}$ partitioned into abstract states $\boldsymbol{A} = \{\boldsymbol{A_i} \mid i <= nAbs\}$ such that sort order $o$ of $\mu(n)$ is maintained across all $\boldsymbol{A_i}$.

1 **begin**
2    **if** $|\boldsymbol{n}| <= nAbs$ **then**
3      $\boldsymbol{A} = \{\{n\} \mid n \in \boldsymbol{n}\}$
4    **else**
5      $\boldsymbol{A} = \Psi_o(\boldsymbol{n}, \mu, nAbs)$
6    **return** $\boldsymbol{A}$
7 **end**

---

# 6 DETAILED DESCRIPTIONS OF ORDERED PARTITIONING SCHEMES FOR VALUE BASED ABSTRACTIONS

We now present seven schemes, each defined by a unique sort order $o$ and partition strategy $\Psi$ combination. Each scheme uses a different method to partition nodes into abstract states keeping the nodes in sort order according to $o$. With a provided value function $\mu(.)$, each scheme can be used to form an ordered value abstraction function. In addition to defining each scheme, we also describe the motivation behind its creation.

**Running Example**   As we motivate and describe the schemes, we will also provide an example of abstract states that would result from partitioning the following nodes:

$$\{1.0, 1.1, 1.2, 1.3, 1.4, 1.5, 10, 100\} \tag{4}$$

into $nAbs = 4$ abstract states by each of partitioning schemes that will be presented.

### 6.0.1   simpleVB

$\Psi_{simpleVB}$ (Algorithm 3)

---

**Algorithm 3:** $\Psi_{simpleVB}$

---

**input**  :A set of nodes $\boldsymbol{n}$ to be partitioned into $nAbs$ abstract states; a value function $\mu(.)$
**output** :$\boldsymbol{n}$ partitioned into abstract states[1] $\boldsymbol{A} = \{\boldsymbol{A_i} \mid i \in \{1, ..., nAbs\}\}$ such that $\forall \boldsymbol{A_i}, \boldsymbol{A_j} \in \boldsymbol{A}, -1 \le |\boldsymbol{A_i}| - |\boldsymbol{A_j}| \le 1$

1 **begin**
2     $baseCardinality \leftarrow \lfloor \frac{|\boldsymbol{n}|}{nAbs} \rfloor$
3     $extras \leftarrow |\boldsymbol{n}| \mod nAbs$
4     $\boldsymbol{n^*} \leftarrow SORT(\boldsymbol{n}, \mu, \text{low-to-high})$
5     $j_{begin} \leftarrow 1$
6     **foreach** $i \leftarrow 1, ..., nAbs$ **do**
7        **if** $extras > 0$ **then**
8           $j_{end} \leftarrow j_{begin} + baseCardinality$
9           $extras \leftarrow extras - 1$
10        **else**
11           $j_{end} \leftarrow j_{begin} + baseCardinality - 1$
12        $\boldsymbol{A_i} \leftarrow \{n^*_{j_{begin}}, ..., n^*_{j_{end}}\}$
13        $j_{begin} \leftarrow j_{end} + 1$
14     **end**
15     $\boldsymbol{A} \leftarrow \cup_{i=1}^{nAbs} \boldsymbol{A_i}$
16     **return** $\boldsymbol{A}$
17 **end**

---

The simpleVB (simple value-based) scheme follows the motivation of grouping nodes of similar value in the same abstract state by a simple 2-step method: 1) first, nodes are ordered by their heuristic value (low to high), and 2) next the ordered nodes are partitioned into [approximately] equal cardinality abstract states.

***Time Complexity.***
Partitioning is achieved via one pass through $|\boldsymbol{n^*}|$ leading to $\mathcal{O}(|\boldsymbol{n}| \log |\boldsymbol{n}|)$ time complexity due to sorting.

***Space Complexity.***
No more than linear space is required. $\mathcal{O}(|\boldsymbol{n}|)$

***Result on Running Example.***
$\{1.0, 1.1\}, \{1.2, 1.3\}, \{1.4, 1.5\}, \{10, 100\}$

Through its simplicity, this method aims to leverage speed allowing for abstractions to be formed much quicker leading to greater number of samples.

---

[1]Such that nodes maintain sort order $o$ across all abstract states.

### 6.0.2 minVarVB

$$\Psi = \Psi_{minVarVB} \text{ (Algorithm 4)}$$

---

**Algorithm 4:** $\Psi_{minVarVB}$

---

**input** : A set of nodes $\boldsymbol{n}$ to be partitioned into $nAbs$ abstract states; a value function $\mu(.)$
**output** : $\boldsymbol{n}$ partitioned into abstract states[1] $\boldsymbol{A} = \{\boldsymbol{A_i} \mid i \in \{1, ..., nAbs\}\}$ satisfying $\min \sum_{\boldsymbol{A_i} \in \boldsymbol{A}} Var(\boldsymbol{A_i}, v)$

1 **begin**
2    $\boldsymbol{A} = WardsMethod(\boldsymbol{n}, nAbs, \mu(\cdot), \text{Euclidian distance})$
3    **return** $\boldsymbol{A}$
4 **end**

---

As mentioned in Section 4.2, Proposition 4.2.0.3, Rizzo [2007] showed that in stratified importance sampling minimizing variance of the estimates within individual strata can lead to a reduction in overall variance.

The minVarVB scheme was designed based on this intuition. The scheme uses Ward's Minimum Variance Hierarchical Clustering (or Ward's Method, for short) Ward [1963] to group nodes into a $nAbs$ abstract states so as to minimize variance within each abstract state with respect to the provided value function $\mu(.)$.

Ward's Minimum Variance Hierarchical Clustering is an agglomerative hierarchical clustering algorithm designed to create a dendrogram by iteratively merging clusters. The primary objective is to minimize the total within-cluster variance. Ward's method works as outlined in Algorithm 5.

---

**Algorithm 5:** Ward's Method

---

1. **Initialization:** Treat each data point as an individual cluster. Assign each cluster a label or identifier.

2. **Compute Pairwise Distances:** Calculate the pairwise distances between all clusters. Various distance metrics can be used, such as Euclidean distance.

3. **Cluster Merging Iteration:**

    (a) Identify the pair of clusters $\boldsymbol{C_i}$ and $\boldsymbol{C_j}$ that, when merged into a new cluster $\boldsymbol{C_{ij}}$, results in the smallest increase in the overall within-cluster variance. This is determined using the formula:
    $$\Delta Var = Var(\boldsymbol{C_{ij}}) - (Var(\boldsymbol{C_i}) + Var(\boldsymbol{C_j}))$$
    where $Var(\boldsymbol{C_{ij}})$ is the variance of the merged cluster, and $Var(\boldsymbol{C_i})$ and $Var(\boldsymbol{C_j})$ are the variances of clusters $\boldsymbol{C_i}$ and $\boldsymbol{C_j}$, respectively.

    (b) Update distance measures between the newly merged cluster and all other clusters.

4. **Repeat:** Repeat steps 2-3 until the desired number of clusters is achieved.

---

Ward's Method can be combined with Lance-Williams linear distance updates Lance and Williams [1967] to increase efficiency. Lance-Williams linear distance updates, in the context of agglomerative clustering, refer to the formula used to calculate the distance between clusters as they are merged during the hierarchical clustering process. The general form of Lance-Williams distance updates can be expressed as follows:

$$d_{(ij)k} = \alpha_i d_{ik} + \alpha_j d_{jk} + \alpha d_{ij} + \gamma |d_{ik} - d_{jk}| \tag{5}$$

where:

- $d_{ij}$, $d_{ik}$, and $d_{jk}$ are the pair-wise distances between clusters $\boldsymbol{C_i}$, $\boldsymbol{C_j}$, and $\boldsymbol{C_k}$

- $d_{(ij)k}$ is the distance between the newly merged cluster $\boldsymbol{C_i} \cup \boldsymbol{C_j}$ and cluster $\boldsymbol{C_k}$

- $\alpha_i, \alpha_j, \alpha,$ and $\gamma$ are coefficients that depend on the linkage criterion used

In the case of Ward's method, the coefficients are specific to the minimization of within-cluster variance and are calculated

as follows:

$$\alpha_i = \frac{|C_i| + |C_k|}{|C_i| + |C_j| + |C_k|}$$

$$\alpha_j = \frac{|C_j| + |C_k|}{|C_i| + |C_j| + |C_k|} \tag{6}$$

$$\alpha = -\frac{|C_k|}{|C_i| + |C_j| + |C_k|}$$

$$\gamma = 0$$

(The inclusion of $\gamma$ provides additional flexibility in the more general case, adjusting the distance updates based on the specific clustering criterion being used).

### Time Complexity.[2]
The choice of clusters to merge generally leads to having a $\mathcal{O}(|\boldsymbol{n}|^3)$ time complexity due to the need to compare pair-wise distances between all clusters at each iteration. However, in the case where nodes are distributed linearly in one dimension, use of a priority queue, and using Lance-Williams distance updates, the time complexity is can be reduced to $\mathcal{O}(|\boldsymbol{n}|^2)$.

### Space Complexity.[2]
The space complexity is implementation dependent, with most time-efficient variants making use of a distance matrix leading to $\mathcal{O}(|\boldsymbol{n}|^2)$ space complexity.

### Result on Running Example.
$\{1.0, 1.1, 1.2\}, \{1.3, 1.4, 1.5\}, \{10\}, \{100\}$

In contrast to simpleVB, minVarVB places considerable resources into computing abstractions, leading to fewer samples, but with potentially better estimates with an appropriate value function $\mu(.)$.

### 6.0.3 equalDistVB

$\Psi_{equalDistVB}$ (Algorithm 6)

---

**Algorithm 6:** $\Psi_{equalDistVB}$

**input** : A set of nodes $\boldsymbol{n}$ to be partitioned into $nAbs$ abstract states; a value function $\mu(.)$

**output** : With $\mu(\boldsymbol{A_{1,...,i}}) = (\sum_{j=1}^{i} \sum_{n' \in \boldsymbol{A_j}} \mu(n')$, $n_{\boldsymbol{A_i}}^{\text{last}}$ be the last node in $\boldsymbol{A_i}$, and $\mathcal{Q}_i = \frac{i \cdot \sum_{n \in \boldsymbol{n}^*} \mu(n)}{nAbs}$, $\boldsymbol{n}$ partitioned into abstract states[1] $\boldsymbol{A} = \{\boldsymbol{A_i} \mid i \in \{1, ..., nAbs\}\}$ such that for $i = 1, ..., nAbs$ in order, $( \mu(\boldsymbol{A_{1,...,i}}) \geq \mathcal{Q}_i ) \wedge ( ( \boldsymbol{A_i} = \{\} ) \vee ( \mu(\boldsymbol{A_{1,...,i}}) - \mu(n_{\boldsymbol{A_i}}^{\text{last}}) < \mathcal{Q}_i ) )$

1 **begin**
2    $\boldsymbol{n}^* \leftarrow SORT(\boldsymbol{n}, \mu, \text{low-to-high})$
3    $j \leftarrow 1$
4    **foreach** $i \leftarrow 1, ..., nAbs$ **do**
5      $\boldsymbol{A_i} \leftarrow \{\}$
6      **while** $\mu(\boldsymbol{A_{1,...,i}}) < \mathcal{Q}_i$ **do**
7        $\boldsymbol{A_i} \leftarrow A_i \cup \{n_j^*\}$
8        $j \leftarrow j + 1$
9      **end**
10    **end**
11    $\boldsymbol{A} \leftarrow \cup_{i=1}^{nAbs} \boldsymbol{A_i}$
12    **return** $\boldsymbol{A}$
13 **end**

---

In sampling it is generally beneficial to predominantly sample high impact regions of the search/sampling space. Allowing the provided value function $\mu(.)$ to serve as a heuristic of nodes that are part of these high impact spaces, equalDistVB attempts to balance this intuition with the notion of variance reduction from minVarVB in attempts to group fewer predicted high impact nodes together in abstract states and allowing for the predicted lower impact nodes to be part of larger abstract states. Also inspired by the simplicity of simpleVB, the scheme works by greedily adding nodes in value order (low to high) into abstract state $\boldsymbol{A_i}$ until the total sum of node values from $\boldsymbol{A_1}, ..., \boldsymbol{A_i}$ reaches or exceeds the $\frac{i}{nAbs}$ quantile.

---

[2]Assuming $\mu(n)$ is $\mathcal{O}(1)$ in both time and space.

When paired with the QB abstraction class, the equalDistVB schemes also attempts to partition notes into abstract states of equal mass under the proposal. This in corresponds to the condition for Proposition 4.2.0.3 for stratified importance sampling variance reduction.

***Time Complexity.[2]***
$\mu(\boldsymbol{A_{1...i}})$ can be updated progressively in constant time, and thus computation of $\mathcal{Q}_i$ at each iteration can also be done in constant time. Partitioning is achieved via one pass through $|\boldsymbol{n^*}|$ leading to $\mathcal{O}(|\boldsymbol{n}|\,log|\boldsymbol{n}|)$ time complexity due to sorting.

***Space Complexity.[2]***
No more than linear space is required. $\mathcal{O}(|\boldsymbol{n}|)$

***Result on Running Example.***
$\{1.0, 1.1, 1.2, 1.3, 1.4, 1.5, 10, 100\}, \{\}, \{\}, \{\}$

Although, this method hopes to find a balance between intuitions previously explored, and without compromising speed and efficiency of abstract state generation, from the running example we can see how this method yield undesirable results in the presence of certain distributions of node values. In this example, the first quantile is only reached after all the nodes have been added to the first abstract state, leaving no nodes remaining to be partitioned into the subsequent abstract states.

### 6.0.4 equalDistVB2

$\Psi_{equalDistVB2}$ (Algorithm 7)

---

**Algorithm 7:** $\Psi_{equalDistVB2}$

---

**input** : A set of nodes $\boldsymbol{n}$ to be partitioned into $nAbs$ abstract states; a value function $\mu(.)$

**output** : With $\mu(\boldsymbol{A_{1,...,i}}) = (\sum_{j=1}^{i} \sum_{n' \in \boldsymbol{A_j}} \mu(n'))$, $n_{\boldsymbol{A_i}}^{\text{last}}$ be the last node in $\boldsymbol{A_i}$, and $\mathcal{Q}_i = \frac{i \cdot \sum_{n \in \boldsymbol{n^*}} \mu(n)}{nAbs}$, $\boldsymbol{n}$ partitioned into abstract states[1] $\boldsymbol{A} = \{\boldsymbol{A_i} \mid i \in \{1, ..., nAbs\}\}$ such that for $i = 1, ..., nAbs$ in order, $(\mu(\boldsymbol{A_{1,...,i}}) \geq \mathcal{Q}_i) \wedge ((\boldsymbol{A_i} = \{\}) \vee (\mu(\boldsymbol{A_{1,...,i}}) - \mu(n_{\boldsymbol{A_i}}^{\text{last}}) < \mathcal{Q}_i))$

1 **begin**
2      $\boldsymbol{n^*} \leftarrow SORT(\boldsymbol{n}, \mu, \text{high-to-low})$
3      $j \leftarrow 1$
4      **foreach** $i \leftarrow 1, ..., nAbs$ **do**
5          $\boldsymbol{A_i} \leftarrow \{\}$
6          **while** $\mu(\boldsymbol{A_{1,...,i}}) < \mathcal{Q}_i$ **do**
7              $\boldsymbol{A_i} \leftarrow \boldsymbol{A_i} \cup \{n_j^*\}$
8              $j \leftarrow j + 1$
9          **end**
10      **end**
11      $\boldsymbol{A} \leftarrow \cup_{i=1}^{nAbs} \boldsymbol{A_i}$
12      **return** $\boldsymbol{A}$
13 **end**

---

By simply reversing the sort order, equalDistVB2 is able to use the same partitioning strategy $\Psi_{equalDistVB}$ associated with equalDistVB meanwhile mitigate some of the overfilling of abstract states.

***Time Complexity.[2]***
$\mu(\boldsymbol{A_{1...i}})$ can be updated progressively in constant time, and thus computation of $\mathcal{Q}_i$ at each iteration can also be done in constant time. Partitioning is achieved via one pass through $|\boldsymbol{n^*}|$ leading to $\mathcal{O}(|\boldsymbol{n}|\,log|\boldsymbol{n}|)$ time complexity due to sorting.

***Space Complexity.[2]***
No more than linear space is required. $\mathcal{O}(|\boldsymbol{n}|)$

***Result on Running Example.***
$\{100\}, \{\}, \{\}, \{10, 1.5, 1.4, 1.3, 1.2, 1.1, 1.0\}$

We see that equalDistVB2 can still be subject to over packing of abstract states. Next we present two more equalDistvB variants that continue to mitigate this artifact.

### 6.0.5 equalDistVB3

$\Psi_{equalDistVB3}$ (Algorithm 8)

---

**Algorithm 8:** $\Psi_{equalDistVB3}$

---

**input** : A set of nodes $\boldsymbol{n}$ to be partitioned into $nAbs$ abstract states; a value function $\mu(.)$

**output** : With $\mu(\boldsymbol{A_{1,...,i}}) = (\sum_{j=1}^{i} \sum_{n' \in \boldsymbol{A_j}} \mu(n'))$, $n_{\boldsymbol{A_i}}^{\text{last}}$ be the last node in $\boldsymbol{A_i}$, and $\mathcal{Q}_i = \frac{i \cdot \sum_{n \in \boldsymbol{n^*}} \mu(n)}{nAbs}$, $\boldsymbol{n}$ partitioned into

    abstract states[1] $\boldsymbol{A} = \{\boldsymbol{A_i} \mid i \in \{1, ..., nAbs\}\}$ such that for $i = 1, ..., nAbs$ in order, $(\mu(\boldsymbol{A_{1,...,i}}) \geq \mathcal{Q}_i) \wedge$

    $((|\boldsymbol{A_i}| = 1) \vee (\mu(\boldsymbol{A_{1,...,i}}) - \mu(n_{\boldsymbol{A_i}}^{\text{last}}) < \mathcal{Q}_i))$

**1 begin**

**2**    $\boldsymbol{n^*} \leftarrow SORT(\boldsymbol{n}, \mu, \text{high-to-low})$

**3**    $j \leftarrow 1$

**4**    **foreach** $i \leftarrow 1, ..., nAbs$ **do**

**5**       $\boldsymbol{A_i} \leftarrow \{n_j^*\}$

**6**       $j \leftarrow j + 1;$

**7**       **while** $\mu(\boldsymbol{A_{1,...,i}}) < \mathcal{Q}_i$ **do**

**8**         $\boldsymbol{A_i} \leftarrow A_i \cup \{n_j^*\}$

**9**         $j \leftarrow j + 1$

**10**      **end**

**11**    **end**

**12**    $\boldsymbol{A} \leftarrow \cup_{i=1}^{nAbs} \boldsymbol{A_i}$

**13**    **return** $\boldsymbol{A}$

**14 end**

---

In order to lessen over packing and ensure abtract states are not left empty, equalDistVB3 modifies equalDistVB2 so that, after processing of each abstract state, the next state is forced an addition of at least a single node by default.

***Time Complexity.[2]***
$\mu(\boldsymbol{A_{1...i}})$ can be updated progressively in constant time, and thus computation of $\mathcal{Q}_i$ at each iteration can also be done in constant time. Partitioning is achieved via one pass through $|\boldsymbol{n^*}|$ leading to $\mathcal{O}(|\boldsymbol{n}| \log|\boldsymbol{n}|)$ time complexity due to sorting.

***Space Complexity.[2]***
No more than linear space is required. $\mathcal{O}(|\boldsymbol{n}|)$

***Result on Running Example.***
$\{100\}, \{10\}, \{1.5\}, \{1.4, 1.3, 1.2, 1.1, 1.0\}$

Still highly efficient, equalDistVB3 manages to ensure that the provided $nAbs$ granularity is honored, allowing users better control of the search vs. sampling interpolation possible with Abstraction Sampling.

### 6.0.6 equalDistVB4

$\Psi_{equalDistVB4}$ (Algorithm 9)

The final varaint of the equalDist schemes, equalDistVB4 attempts to perform a more even partitioning than the previous variants by recomputing quantiles. Each time the algorithm progesses to processing a new abstract state, remaining nodes and abstract states are used to compute new quantiles which are then used to guide filling of the current abstract state in the same way previously done.

***Time Complexity.[2]***
$\mu(\boldsymbol{A_{1...i}})$ can be updated progressively in constant time, and thus computation of $\widehat{\mathcal{Q}}_i$ at each iteration can also be done in constant time. Partitioning is achieved via one pass through $|\boldsymbol{n^*}|$ leading to $\mathcal{O}(|\boldsymbol{n}| \log|\boldsymbol{n}|)$ time complexity due to sorting.

***Space Complexity.[2]***
No more than linear space is required. $\mathcal{O}(|\boldsymbol{n}|)$

***Result on Running Example.***
$\{100\}, \{10\}, \{1.5, 1.4, 1.3\}, \{1.2, 1.1, 1.0\}$

Still highly efficient, equalDistVB3 manages to ensure that the provided $nAbs$ granularity is honored, allowing users better

**Algorithm 9:** $\Psi_{equalDistVB4}$

---

**input** : A set of nodes $\boldsymbol{n}$ to be partitioned into $nAbs$ abstract states; a value function $\mu(.)$

**output** : With $\mu(\boldsymbol{A_{1,...,i}}) = (\sum_{j=1}^{i} \sum_{n' \in \boldsymbol{A_j}} \mu(n'))$, $n_{\boldsymbol{A_i}}^{\text{last}}$ be the last node in $\boldsymbol{A_i}$, and $\widehat{\mathcal{Q}}_i = \frac{\mu(\boldsymbol{n^*}) - \mu(\boldsymbol{A_{1,...,i-1}})}{nAbs - i + 1}$, $\boldsymbol{n}$ partitioned

into abstract states[1] $\boldsymbol{A} = \{\boldsymbol{A_i} \mid i \in \{1, ..., nAbs\}\}$ such that for $i = 1, ..., nAbs$ in order, ( $\mu(\boldsymbol{A_i}) \geq \widehat{\mathcal{Q}}_i$ ) $\wedge$

( ( $|\boldsymbol{A_i}| = 1$ ) $\vee$ ( $\mu(\boldsymbol{A_i}) - \mu(n_{\boldsymbol{A_i}}^{\text{last}}) < \widehat{\mathcal{Q}}_i$ ) )

**1 begin**

**2**    $\boldsymbol{n^*} \leftarrow SORT(\boldsymbol{n}, \mu, \text{high-to-low})$

**3**    $j \leftarrow 1$

**4**    **foreach** $i \leftarrow 1, ..., nAbs$ **do**

**5**      $\boldsymbol{A_i} \leftarrow \{\}$

**6**      **while** $\mu(\boldsymbol{A_i}) < \widehat{\mathcal{Q}}_i$ **do**

**7**        $\boldsymbol{A_i} \leftarrow A_i \cup \{n_j^*\}$

**8**        $j \leftarrow j + 1$

**9**      **end**

**10**    **end**

**11**    $\boldsymbol{A} \leftarrow \cup_{i=1}^{nAbs} \boldsymbol{A_i}$

**12**    **return** $\boldsymbol{A}$

**13 end**

---

control of the search vs. sampling interpolation possible with Abstraction Sampling.

### 6.0.7 randVB

$\Psi_{randVB}$ (Algorithm 10)

**Algorithm 10:** $\Psi_{randVB}$

---

**input** : A set of nodes $\boldsymbol{n}$ to be partitioned into $nAbs$ abstract states; a value function $\mu(.)$

**output** : $\boldsymbol{n}$ partitioned into abstract states[1] $\boldsymbol{A} = \{\boldsymbol{A_i} \mid i \in \{1, ..., nAbs\}\}$

**1 begin**

**2**    $\boldsymbol{n^*} \leftarrow SORT(\boldsymbol{n}, \mu, \text{high-to-low})$

**3**    $\boldsymbol{s} \sim Unif(\{\boldsymbol{M} \subseteq \{1, ..., |\boldsymbol{n^*}| - 1\} \mid |\boldsymbol{M}| = nAbs - 1\})$

**4**    $\boldsymbol{s^*} \leftarrow SORT(\boldsymbol{s})$

**5**    $j \leftarrow 1$

**6**    **foreach** $i \leftarrow 1, ..., nAbs - 1$ **do**

**7**      $\boldsymbol{A_i} = \{n_j^*, ..., n_{s_i^*}^*\}$

**8**      $j \leftarrow s_i^* + 1$

**9**    **end**

**10**    $\boldsymbol{A_{nAbs}} = \{n_j^*, ..., n_{|\boldsymbol{n^*}|}^*\}$

**11**    $\boldsymbol{A} = \cup_{i=1}^{nAbs} \boldsymbol{A_i}$

**12**    **return** $\boldsymbol{A}$

**13 end**

---

If the quality of $\mu(.)$ as a measure of similarity is unknown or poor, it could instead be beneficial to rely on randomness to ensure a diverse sampling of abstractions. randVB does this by sampling $nAbs - 1$ partition points between the sorted nodes $\boldsymbol{n^*}$ uniformly at random and without replacement, and then partitions the nodes accordingly. As a result, abstract states are formed such that nodes are still grouped according to $\mu(.)$, but the size of those groups varies.

***Time Complexity.[2]***
$\mathcal{O}(|\boldsymbol{n}| \log |\boldsymbol{n}|)$ time complexity due to sorting.

***Space Complexity.[2]***
No more than linear space is required. $\mathcal{O}(|\boldsymbol{n}|)$

***Result on Running Example.***
$\{100, 10\}, \{1.5\}, \{1.4, 1.3, 1.2\}, \{1.1, 1.0\};$
$\{100\}, \{10, 1.5, 1.4, 1.3\}, \{1.2, 1.1\}, \{1.0\};$ ...etc.

# 7 EXTENDED RESULTS

In extension to the main paper, here we show a more comprehensive set of aggregated data tables, now also including the standard deviation of the errors, the average number of samples drawn, and average probe sizes.

## 7.1 SUMMARY COMPARISON.

### 7.1.1 Exact Problems

| iB-5, t-1300sec, Exact | | | | DBN | | | |
|---|---|---|---|---|---|---|---|
| Class | Scheme | nAbs | Fail | Avg. Error | std(Avg. Error) | Avg. Num. Samples | Avg. Probe Size |
| HB | simple | 2048 | 0 | 0.440 | 0.862 | 354 | 233936 |
| | minVar | 1 | 0 | 1.361 | 2.840 | 600260 | 136 |
| | equalDist | 1 | 0 | 1.365 | 2.835 | 634640 | 136 |
| | equalDist2 | 1 | 0 | 1.570 | 3.292 | 493719 | 196 |
| | equalDist3 | 1 | 0 | 1.489 | 3.018 | 489934 | 196 |
| | equalDist4 | 1024 | 0 | 2.819 | 5.501 | 114 | 2965761 |
| | rand | 256 | 0 | 0.496 | 0.796 | 2840 | 30952 |
| HRB | simple | 2048 | 0 | 0.491 | 0.976 | 353 | 233936 |
| | minVar | 1 | 0 | 1.500 | 2.972 | 635538 | 136 |
| | equalDist | 1 | 0 | 1.305 | 2.508 | 654598 | 136 |
| | equalDist2 | 1 | 0 | 1.549 | 3.405 | 664595 | 136 |
| | equalDist3 | 1 | 0 | 1.405 | 3.014 | 662702 | 136 |
| | equalDist4 | 1 | 0 | 1.511 | 3.064 | 664347 | 136 |
| | rand | 2048 | 0 | 0.451 | 0.719 | 358 | 233936 |
| QB | simple | 1 | 0 | 1.469 | 2.920 | 677854 | 136 |
| | minVar | 2048 | 0 | 0.050 | 0.173 | 10 | 233936 |
| | equalDist | 4 | 0 | 1.174 | 2.407 | 478845 | 181 |
| | equalDist2 | 2048 | 0 | 0.736 | 1.831 | 17787 | 3326 |
| | equalDist3 | 2048 | 0 | 0.042 | 0.137 | 346 | 233936 |
| | equalDist4 | 2048 | 0 | 0.130 | 0.378 | 1969 | 153490 |
| | rand | 1 | 0 | 1.295 | 2.723 | 683431 | 136 |
| CTX | rand | 4 | 0 | 1.381 | 2.626 | 197143 | 476 |
| | rel | 1 | 0 | 1.472 | 3.093 | 695636 | 136 |
| RAND | rand | 2048 | 0 | 0.104 | 0.243 | 359 | 233936 |

**Table 1**

| iB-5, t-300sec, Exact | | | | Grids | | | |
|---|---|---|---|---|---|---|---|
| Class | Scheme | nAbs | Fail | Avg. Error | std(Avg. Error) | Avg. Num. Samples | Avg. Probe Size |
| HB | simple | 1024 | 0 | 2.202 | 3.807 | 1536 | 365339 |
| | minVar | 16 | 0 | 3.251 | 5.615 | 37401 | 6295 |
| | equalDist | 2048 | 0 | 10.854 | 19.810 | 12787 | 36088 |
| | equalDist2 | 512 | 0 | 8.050 | 14.709 | 44538 | 11654 |
| | equalDist3 | 2048 | 0 | 2.764 | 4.210 | 588 | 805429 |
| | equalDist4 | 64 | 0 | 6.029 | 11.585 | 10521 | 359937 |
| | rand | 2048 | 0 | 2.248 | 3.933 | 709 | 737966 |
| HRB | simple | 4 | 0 | 9.667 | 17.275 | 441504 | 1678 |
| | minVar | 64 | 0 | 2.319 | 3.816 | 3046 | 25570 |
| | equalDist | 256 | 0 | 10.635 | 18.892 | 86568 | 6357 |
| | equalDist2 | 2048 | 0 | 6.790 | 11.752 | 12056 | 35124 |
| | equalDist3 | 1024 | 0 | 2.292 | 3.951 | 1259 | 396048 |
| | equalDist4 | 512 | 0 | 1.829 | 3.057 | 2787 | 188320 |
| | rand | 4 | 0 | 6.122 | 10.479 | 465813 | 1643 |
| QB | simple | 16 | 0 | 10.076 | 17.905 | 113719 | 6499 |
| | minVar | 1024 | 0 | 1.566 | 2.844 | 14 | 397296 |
| | equalDist | 2048 | 0 | 8.134 | 16.643 | 12162 | 70457 |
| | equalDist2 | 2048 | 0 | 4.405 | 9.051 | 11932 | 71415 |
| | equalDist3 | 2048 | 0 | 1.771 | 3.391 | 612 | 788719 |
| | equalDist4 | 512 | 0 | 1.754 | 3.159 | 2793 | 190568 |
| | rand | 256 | 0 | 6.048 | 10.294 | 6041 | 100691 |
| CTX | rand | 4 | 0 | 5.030 | 9.168 | 471163 | 1421 |
| | rel | 64 | 0 | 4.021 | 7.528 | 36934 | 14867 |
| RAND | rand | 1024 | 0 | 1.501 | 2.530 | 1504 | 390548 |

**Table 2**

**Pedigree**

| iB-5, t-300sec, Exact | | | | | | | |
|---|---|---|---|---|---|---|---|
| Class | Scheme | nAbs | Fail | Avg. Error | std(Avg. Error) | Avg. Num. Samples | Avg. Probe Size |
| HB | simple | 2048 | 0 | 0.150 | 0.564 | 393 | 1208067 |
| | minVar | 64 | 0 | 0.422 | 0.894 | 1904 | 34760 |
| | equalDist | 1024 | 0 | 0.303 | 0.626 | 1104 | 406884 |
| | equalDist2 | 1024 | 0 | 0.315 | 0.536 | 1090 | 410306 |
| | equalDist3 | 1024 | 0 | 0.279 | 0.539 | 727 | 606552 |
| | equalDist4 | 512 | 0 | 0.214 | 0.622 | 1526 | 305759 |
| | rand | 2048 | 0 | 0.185 | 0.473 | 406 | 1170793 |
| HRB | simple | 256 | 0 | 0.225 | 0.378 | 3637 | 155656 |
| | minVar | 256 | 0 | 0.309 | 0.543 | 131 | 149534 |
| | equalDist | 1024 | 0 | 0.638 | 0.921 | 1653 | 247759 |
| | equalDist2 | 16 | 0 | 0.457 | 0.646 | 83869 | 5396 |
| | equalDist3 | 16 | 0 | 0.537 | 0.843 | 63832 | 8067 |
| | equalDist4 | 64 | 0 | 0.483 | 0.836 | 14789 | 34813 |
| | rand | 64 | 0 | 0.666 | 0.983 | 17216 | 36226 |
| QB | simple | 256 | 0 | 0.297 | 0.510 | 3672 | 153687 |
| | minVar | 64 | 0 | 0.210 | 0.561 | 1939 | 36977 |
| | equalDist | 2048 | 0 | 0.144 | 0.646 | 524 | 808760 |
| | equalDist2 | 1024 | 0 | 0.145 | 0.637 | 1067 | 410631 |
| | equalDist3 | 512 | 0 | 0.148 | 0.643 | 1403 | 324983 |
| | equalDist4 | 512 | 0 | 0.134 | 0.600 | 1415 | 322792 |
| | rand | 16 | 0 | 0.740 | 1.021 | 76974 | 8055 |
| CTX | rand | 16 | 0 | 0.540 | 0.827 | 169911 | 2790 |
| | rel | 64 | 0 | 0.424 | 0.653 | 28214 | 29061 |
| RAND | rand | 1024 | 0 | 0.143 | 0.619 | 878 | 620063 |

**Table 3**

**Promedas**

| iB-5, t-300sec, Exact | | | | | | | |
|---|---|---|---|---|---|---|---|
| Class | Scheme | nAbs | Fail | Avg. Error | std(Avg. Error) | Avg. Num. Samples | Avg. Probe Size |
| HB | simple | 1024 | 0 | 0.575 | 1.288 | 7878163 | 215898 |
| | minVar | 16 | 2 | 2.509 | 5.329 | 4119191 | 2304 |
| | equalDist | 1024 | 0 | 2.332 | 3.857 | 8057221 | 145513 |
| | equalDist2 | 64 | 0 | 2.123 | 4.632 | 8086745 | 8209 |
| | equalDist3 | 256 | 0 | 2.196 | 4.354 | 8212578 | 53287 |
| | equalDist4 | 2048 | 0 | 1.355 | 2.486 | 8106429 | 471471 |
| | rand | 2048 | 0 | 0.752 | 1.476 | 8136226 | 382946 |
| HRB | simple | 2048 | 0 | 0.705 | 1.594 | 8281435 | 444640 |
| | minVar | 16 | 1 | 2.801 | 5.552 | 8302630 | 2403 |
| | equalDist | 16 | 4 | 4.055 | 7.212 | 8505442 | 1255 |
| | equalDist2 | 16 | 2 | 3.445 | 6.549 | 8445561 | 1667 |
| | equalDist3 | 16 | 2 | 2.656 | 5.561 | 8389700 | 2330 |
| | equalDist4 | 2048 | 0 | 2.024 | 3.247 | 8278922 | 429451 |
| | rand | 1024 | 1 | 2.165 | 4.691 | 8284836 | 184056 |
| QB | simple | 256 | 1 | 3.164 | 5.634 | 8156519 | 44804 |
| | minVar | 64 | 1 | 1.062 | 3.999 | 8149950 | 13097 |
| | equalDist | 2048 | 0 | 0.583 | 1.053 | 8159447 | 85975 |
| | equalDist2 | 2048 | 0 | 0.539 | 1.098 | 8146812 | 87006 |
| | equalDist3 | 2048 | 0 | 0.412 | 0.917 | 8136397 | 517395 |
| | equalDist4 | 512 | 0 | 0.437 | 1.062 | 8155880 | 126503 |
| | rand | 16 | 2 | 5.988 | 12.148 | 8401169 | 1892 |
| CTX | rand | 1024 | 1 | 2.442 | 4.755 | 8045093 | 2016 |
| | rel | 64 | 6 | 4.349 | 7.852 | 8384108 | 3268 |
| RAND | rand | 1024 | 0 | 0.513 | 1.033 | 8047804 | 228960 |

**Table 4**

## 7.1.2 LARGE Problems

**Table 5**

| iB-10, t-1200sec, LARGE | | | | | | | |
|---|---|---|---|---|---|---|---|
| Class | Scheme | nAbs | Fail | Avg. Error | std(Avg. Error) | Avg. Num. Samples | Avg. Probe Size |
| HB | simple | 512 | 0 | 3.059 | 5.994 | 466 | 213236 |
| | minVar | 1 | 0 | 6.372 | 10.410 | 170425 | 434 |
| | equalDist | 1 | 0 | 6.354 | 10.259 | 171742 | 434 |
| | equalDist2 | 1 | 0 | 6.172 | 9.889 | 146566 | 598 |
| | equalDist3 | 1 | 0 | 6.548 | 10.206 | 144646 | 598 |
| | equalDist4 | 1 | 0 | 6.525 | 10.576 | 162296 | 434 |
| | rand | 64 | 0 | 1.855 | 2.986 | 3682 | 27039 |
| HRB | simple | 2048 | 0 | 3.202 | 6.388 | 116 | 844724 |
| | minVar | 1 | 0 | 6.102 | 9.811 | 167382 | 434 |
| | equalDist | 1 | 0 | 6.273 | 10.219 | 165303 | 434 |
| | equalDist2 | 1 | 0 | 6.689 | 10.719 | 164615 | 434 |
| | equalDist3 | 1 | 0 | 6.564 | 10.301 | 163186 | 434 |
| | equalDist4 | 1 | 0 | 6.606 | 10.704 | 162441 | 434 |
| | rand | 2048 | 0 | 1.915 | 3.994 | 116 | 844724 |
| QB | simple | 1 | 0 | 6.540 | 10.583 | 162844 | 434 |
| | minVar | 2048 | 0 | 1.837 | 4.023 | 11 | 844724 |
| | equalDist | 512 | 0 | 5.423 | 9.545 | 28518 | 50129 |
| | equalDist2 | 2048 | 0 | 3.813 | 7.105 | 11104 | 162286 |
| | equalDist3 | 2048 | 0 | 1.645 | 3.853 | 115 | 844724 |
| | equalDist4 | 2048 | 0 | 1.643 | 3.847 | 170 | 758313 |
| | rand | 4 | 0 | 6.292 | 9.781 | 52602 | 1721 |
| CTX | rand | 64 | 0 | 5.710 | 8.760 | 4947 | 22519 |
| | rel | 1 | 0 | 6.267 | 10.128 | 165870 | 434 |
| RAND | rand | 2048 | 0 | 2.123 | 4.214 | 116 | 844724 |

DBN

**Table 5**

| iB-10, t-1200sec, LARGE | | | | | | | |
|---|---|---|---|---|---|---|---|
| Class | Scheme | nAbs | Fail | Avg. Error | std(Avg. Error) | Avg. Num. Samples | Avg. Probe Size |
| HB | simple | 2048 | 0 | 73.710 | 117.967 | 585 | 5281698 |
| | minVar | 64 | 0 | 71.628 | 112.070 | 1948 | 184817 |
| | equalDist | 2048 | 0 | 149.888 | 252.503 | 6894 | 438547 |
| | equalDist2 | 1024 | 0 | 119.823 | 195.442 | 12859 | 247710 |
| | equalDist3 | 64 | 0 | 82.927 | 124.857 | 15758 | 197893 |
| | equalDist4 | 1024 | 0 | 63.194 | 97.515 | 1020 | 2846847 |
| | rand | 2048 | 0 | 82.203 | 132.286 | 527 | 5492402 |
| HRB | simple | 1024 | 0 | 193.654 | 311.138 | 1042 | 3061184 |
| | minVar | 512 | 0 | 37.972 | 56.653 | 29 | 1534848 |
| | equalDist | 2048 | 0 | 127.990 | 216.992 | 6524 | 475696 |
| | equalDist2 | 2048 | 0 | 104.502 | 168.754 | 6388 | 501514 |
| | equalDist3 | 2048 | 0 | 38.936 | 52.976 | 429 | 6090687 |
| | equalDist4 | 2048 | 0 | 34.676 | 50.051 | 460 | 5664129 |
| | rand | 16 | 0 | 160.168 | 262.678 | 78263 | 48729 |
| QB | simple | 16 | 0 | 197.931 | 331.349 | 73034 | 51032 |
| | minVar | 1024 | 0 | 28.423 | 44.701 | 7 | 3064517 |
| | equalDist | 2048 | 0 | 118.547 | 209.112 | 6013 | 932447 |
| | equalDist2 | 2048 | 0 | 91.994 | 160.979 | 5935 | 939064 |
| | equalDist3 | 2048 | 0 | 19.277 | 31.795 | 429 | 6135039 |
| | equalDist4 | 2048 | 0 | 18.866 | 34.470 | 462 | 5658527 |
| | rand | 16 | 0 | 163.973 | 270.397 | 78137 | 48849 |
| CTX | rand | 512 | 0 | 111.104 | 189.309 | 53385 | 66495 |
| | rel | 1024 | 0 | 80.633 | 131.304 | 1990 | 1210381 |
| RAND | rand | 2048 | 0 | 19.053 | 30.561 | 517 | 5915471 |

Grids

**Table 6**

**Linkage-Type4**

| iB-10, t-1200sec, LARGE | | | | | | | |
|---|---|---|---|---|---|---|---|
| Class | Scheme | nAbs | Fail | Avg. Error | std(Avg. Error) | Avg. Num. Samples | Avg. Probe Size |
| HB | simple | 2048 | 21 | 47.383 | 124.818 | 215 | 6362535 |
| | minVar | 256 | 37 | 133.377 | 208.955 | 118 | 526228 |
| | equalDist | 2048 | 37 | 136.462 | 209.952 | 806 | 1725651 |
| | equalDist2 | 2048 | 36 | 132.531 | 206.716 | 775 | 1763041 |
| | equalDist3 | 2048 | 32 | 118.653 | 191.472 | 373 | 4231305 |
| | equalDist4 | 2048 | 29 | 98.222 | 180.908 | 260 | 5348049 |
| | rand | 2048 | 21 | 52.171 | 117.178 | 258 | 5548243 |
| HRB | simple | 2048 | 17 | 48.474 | 105.528 | 201 | 7170175 |
| | minVar | 512 | 38 | 138.131 | 211.272 | 31 | 1203151 |
| | equalDist | 2048 | 32 | 123.253 | 192.089 | 1138 | 1438777 |
| | equalDist2 | 2048 | 34 | 129.751 | 198.056 | 1114 | 1453506 |
| | equalDist3 | 2048 | 31 | 118.091 | 185.967 | 405 | 4586845 |
| | equalDist4 | 2048 | 26 | 95.895 | 158.305 | 335 | 5182785 |
| | rand | 1024 | 18 | 127.021 | 162.172 | 576 | 3170049 |
| QB | simple | 2048 | 13 | 48.681 | 102.256 | 165 | 7582217 |
| | minVar | 256 | 31 | 93.058 | 176.650 | 115 | 595380 |
| | equalDist | 2048 | 22 | 46.196 | 128.408 | 324 | 3606296 |
| | equalDist2 | 1024 | 21 | 40.310 | 115.108 | 823 | 1613744 |
| | equalDist3 | 1024 | 20 | 37.490 | 115.666 | 428 | 3151667 |
| | equalDist4 | 2048 | 16 | 30.512 | 104.300 | 155 | 7276760 |
| | rand | 256 | 17 | 156.992 | 197.622 | 2123 | 786014 |
| CTX | rand | 2048 | 53 | 194.741 | 250.879 | 78237 | 12693 |
| | rel | 1024 | 37 | 129.189 | 210.249 | 911 | 2128473 |
| RAND | rand | 1024 | 19 | 33.804 | 107.942 | 531 | 3043774 |

**Table 7**

**Promedas**

| iB-10, t-1200sec, LARGE | | | | | | | |
|---|---|---|---|---|---|---|---|
| Class | Scheme | nAbs | Fail | Avg. Error | std(Avg. Error) | Avg. Num. Samples | Avg. Probe Size |
| HB | simple | 1024 | 16 | 5.981 | 14.402 | 5303 | 316842 |
| | minVar | 16 | 25 | 9.433 | 17.375 | 135360 | 3961 |
| | equalDist | 64 | 23 | 9.664 | 16.936 | 122333 | 11729 |
| | equalDist2 | 16 | 22 | 9.465 | 17.026 | 438953 | 3209 |
| | equalDist3 | 16 | 18 | 8.534 | 16.129 | 364644 | 3961 |
| | equalDist4 | 16 | 19 | 8.011 | 15.663 | 368986 | 3973 |
| | rand | 64 | 22 | 8.296 | 16.348 | 129906 | 14836 |
| HRB | simple | 512 | 15 | 5.849 | 14.157 | 10763 | 158416 |
| | minVar | 16 | 24 | 9.577 | 17.048 | 130796 | 4001 |
| | equalDist | 16 | 32 | 11.596 | 19.010 | 546356 | 2629 |
| | equalDist2 | 4 | 25 | 10.380 | 17.881 | 1755156 | 841 |
| | equalDist3 | 16 | 22 | 9.779 | 17.253 | 388573 | 3844 |
| | equalDist4 | 16 | 22 | 9.217 | 16.843 | 383539 | 3876 |
| | rand | 64 | 27 | 9.556 | 17.661 | 128420 | 15010 |
| QB | simple | 4 | 34 | 11.919 | 19.156 | 2214241 | 849 |
| | minVar | 16 | 13 | 5.403 | 13.076 | 127451 | 4261 |
| | equalDist | 512 | 15 | 5.960 | 13.509 | 21151 | 61005 |
| | equalDist2 | 2048 | 12 | 4.982 | 12.955 | 5495 | 230190 |
| | equalDist3 | 256 | 5 | 2.560 | 8.629 | 16078 | 90936 |
| | equalDist4 | 512 | 5 | 2.476 | 8.229 | 7638 | 187975 |
| | rand | 4 | 28 | 11.532 | 19.413 | 2330332 | 841 |
| CTX | rand | 256 | 0 | 3.222 | 5.085 | 160087 | 12862 |
| | rel | 16 | 34 | 11.247 | 18.992 | 761684 | 2399 |
| RAND | rand | 1024 | 10 | 3.936 | 11.615 | 5010 | 348002 |

**Table 8**

## 7.2 COMPARISON USING 100 SAMPLES.

### 7.2.1 Exact Problems

**Table 9**

| iB-5, m-100, Exact | | | DBN | | Grids | | Pedigree | | Promedas | |
|---|---|---|---|---|---|---|---|---|---|---|
| Class | Scheme | nAbs | Fail | Avg. Error | Fail | Avg. Error | Fail | Avg. Error | Fail | Avg. Error |
| HB | simpleQB | 256 | 0 | 1.601 | 0 | 4.768 | 0 | 0.337 | 14 | 3.121 |
|  | minVarQB | 256 | 0 | 5.028 | 0 | 5.134 | 0 | 0.615 | 1 | 5.423 |
|  | equalDist | 256 | 0 | 5.269 | 0 | 15.958 | 1 | 2.145 | 13 | 6.556 |
|  | equalDist2 | 256 | 0 | 5.966 | 0 | 11.009 | 0 | 1.384 | 6 | 6.464 |
|  | equalDist3 | 256 | 0 | 6.203 | 0 | 5.804 | 0 | 0.669 | 1 | 5.480 |
|  | equalDist4 | 256 | 0 | 4.501 | 0 | 22.576 | 0 | 1.103 | 1 | 4.382 |
|  | randQB | 256 | 0 | 0.712 | 0 | 5.515 | 0 | 0.531 | 13 | 4.988 |
| HRB | simpleQB | 256 | 0 | 1.638 | 0 | 15.757 | 0 | 0.721 | 14 | 3.014 |
|  | minVarQB | 256 | 0 | 4.703 | 0 | 2.404 | 0 | 0.287 | 1 | 4.295 |
|  | equalDist | 256 | 0 | 6.030 | 0 | 16.132 | 1 | 2.817 | 13 | 8.830 |
|  | equalDist2 | 256 | 0 | 6.361 | 0 | 10.462 | 0 | 2.546 | 6 | 8.272 |
|  | equalDist3 | 256 | 0 | 6.613 | 0 | 4.236 | 0 | 2.291 | 1 | 7.427 |
|  | equalDist4 | 256 | 0 | 6.753 | 0 | 3.179 | 0 | 1.241 | 1 | 5.552 |
|  | randQB | 256 | 0 | 0.720 | 0 | 9.838 | 0 | 1.818 | 13 | 7.074 |
| QB | simpleQB | 256 | 0 | 5.350 | 0 | 17.406 | 0 | 1.059 | 14 | 9.659 |
|  | minVarQB | 256 | 0 | 0.111 | 0 | 1.911 | 0 | 0.223 | 1 | 1.634 |
|  | equalDist | 256 | 0 | 5.619 | 0 | 15.533 | 1 | 0.858 | 13 | 5.420 |
|  | equalDist2 | 256 | 0 | 2.319 | 0 | 11.220 | 0 | 0.563 | 6 | 3.479 |
|  | equalDist3 | 256 | 0 | 0.173 | 0 | 3.615 | 0 | 0.206 | 1 | 1.473 |
|  | equalDist4 | 256 | 0 | 0.277 | 0 | 2.305 | 0 | 0.180 | 1 | 1.373 |
|  | randQB | 256 | 0 | 4.982 | 0 | 12.653 | 0 | 3.211 | 13 | 19.441 |
| CTX | rand | 256 | 0 | 3.587 | 0 | 9.568 | 2 | 4.695 | 3 | 14.386 |
|  | rel | 256 | 0 | 5.265 | 0 | 8.013 | 0 | 1.097 | 36 | 10.845 |
| RAND | rand | 256 | 0 | 0.288 | 0 | 2.464 | 0 | 0.325 | 3 | 2.570 |

### 7.2.2 LARGE Problems

**Table 10**

| iB-10, m-100, LARGE | | | DBN | | Grids | | Linkage-Type4 | | Promedas | |
|---|---|---|---|---|---|---|---|---|---|---|
| Class | Scheme | nAbs | Fail | Avg. Error | Fail | Avg. Error | Fail | Avg. Error | Fail | Avg. Error |
| HB | simpleQB | 256 | 0 | 4.179 | 0 | 108.953 | 0 | 189.141 | 14 | 16.389 |
|  | minVarQB | 256 | 0 | 8.219 | 0 | 45.460 | 0 | 182.791 | 1 | 17.221 |
|  | equalDist | 256 | 0 | 8.013 | 0 | 164.767 | 1 | 230.627 | 13 | 19.890 |
|  | equalDist2 | 256 | 0 | 8.233 | 0 | 119.203 | 0 | 231.620 | 6 | 18.944 |
|  | equalDist3 | 256 | 0 | 7.905 | 0 | 67.626 | 0 | 219.364 | 1 | 18.612 |
|  | equalDist4 | 256 | 0 | 7.588 | 0 | 54.643 | 0 | 199.565 | 1 | 17.186 |
|  | randQB | 256 | 0 | 3.741 | 0 | 108.760 | 0 | 203.436 | 13 | 18.494 |
| HRB | simpleQB | 256 | 0 | 4.203 | 0 | 190.126 | 0 | 180.424 | 14 | 15.857 |
|  | minVarQB | 256 | 0 | 7.770 | 0 | 29.575 | 0 | 188.654 | 1 | 17.492 |
|  | equalDist | 256 | 0 | 7.947 | 0 | 151.765 | 1 | 235.331 | 13 | 20.390 |
|  | equalDist2 | 256 | 0 | 8.616 | 0 | 114.215 | 0 | 229.609 | 6 | 20.395 |
|  | equalDist3 | 256 | 0 | 7.653 | 0 | 37.005 | 0 | 222.866 | 1 | 19.932 |
|  | equalDist4 | 256 | 0 | 8.201 | 0 | 31.368 | 0 | 213.918 | 1 | 18.694 |
|  | randQB | 256 | 0 | 3.254 | 0 | 150.130 | 0 | 205.219 | 13 | 19.157 |
| QB | simpleQB | 256 | 0 | 7.921 | 0 | 194.220 | 0 | 180.487 | 14 | 22.732 |
|  | minVarQB | 256 | 0 | 2.848 | 0 | 22.838 | 0 | 182.296 | 1 | 11.742 |
|  | equalDist | 256 | 0 | 6.443 | 0 | 140.283 | 1 | 192.449 | 13 | 17.245 |
|  | equalDist2 | 256 | 0 | 4.583 | 0 | 96.859 | 0 | 193.109 | 6 | 15.704 |
|  | equalDist3 | 256 | 0 | 3.036 | 0 | 25.042 | 0 | 170.706 | 1 | 11.426 |
|  | equalDist4 | 256 | 0 | 2.715 | 0 | 20.978 | 0 | 162.793 | 1 | 11.885 |
|  | randQB | 256 | 0 | 7.791 | 0 | 163.214 | 0 | 205.186 | 13 | 23.984 |
| CTX | rand | 256 | 0 | 4.789 | 0 | 97.951 | 2 | 232.778 | 3 | 16.285 |
|  | rel | 256 | 0 | 7.664 | 0 | 65.146 | 0 | 188.194 | 36 | 20.609 |
| RAND | rand | 256 | 0 | 3.070 | 0 | 26.185 | 0 | 178.273 | 3 | 13.957 |

## 7.3 TIME SERIES PLOT

### 7.3.1 LARGE Problems

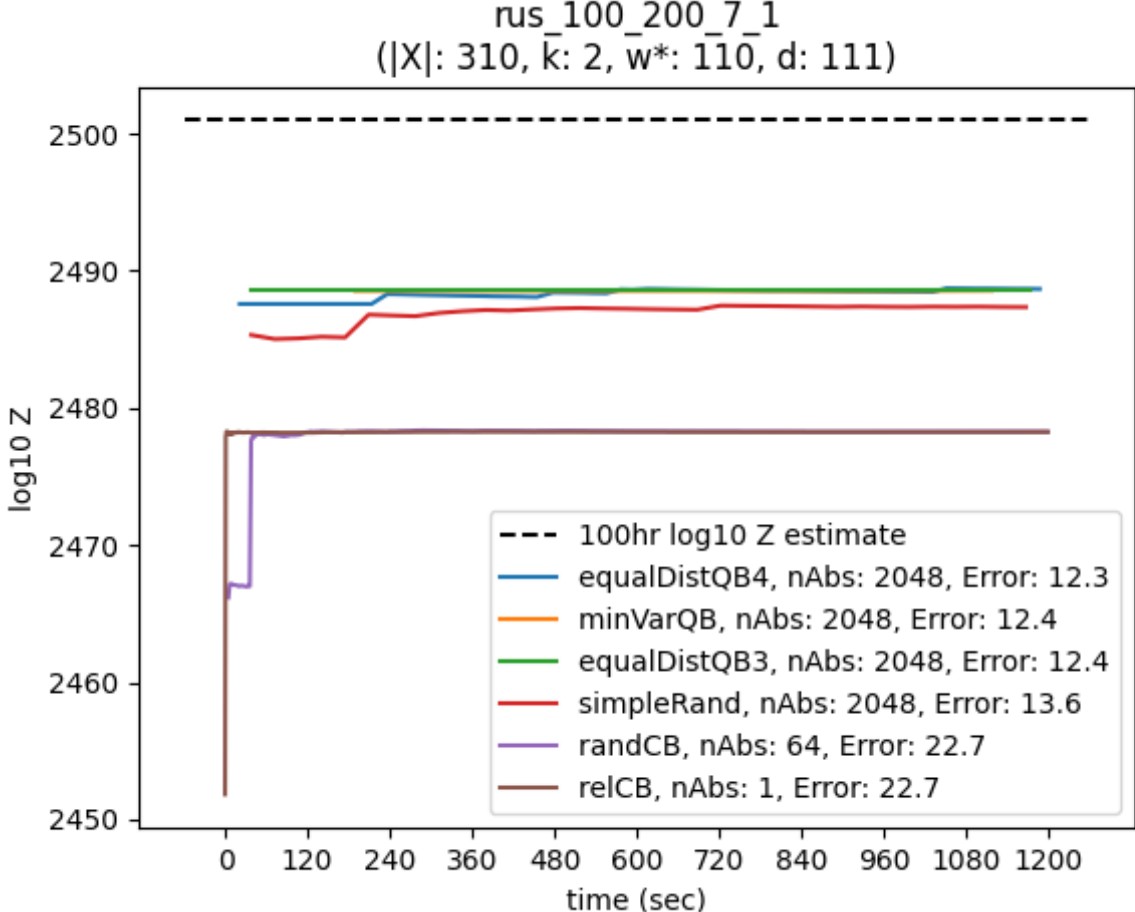

**Plot 1:** Z estimates from various algorithms versus time on DBN problem rus_100_200_7_1 using $iB = 10$. The dashed black line shows the estimated true Z value.

**Plot 2:** Z estimates from various algorithms versus time on Grids problem grid80x80.f10.wrap using $iB = 10$. The dashed black line shows the estimated true Z value.

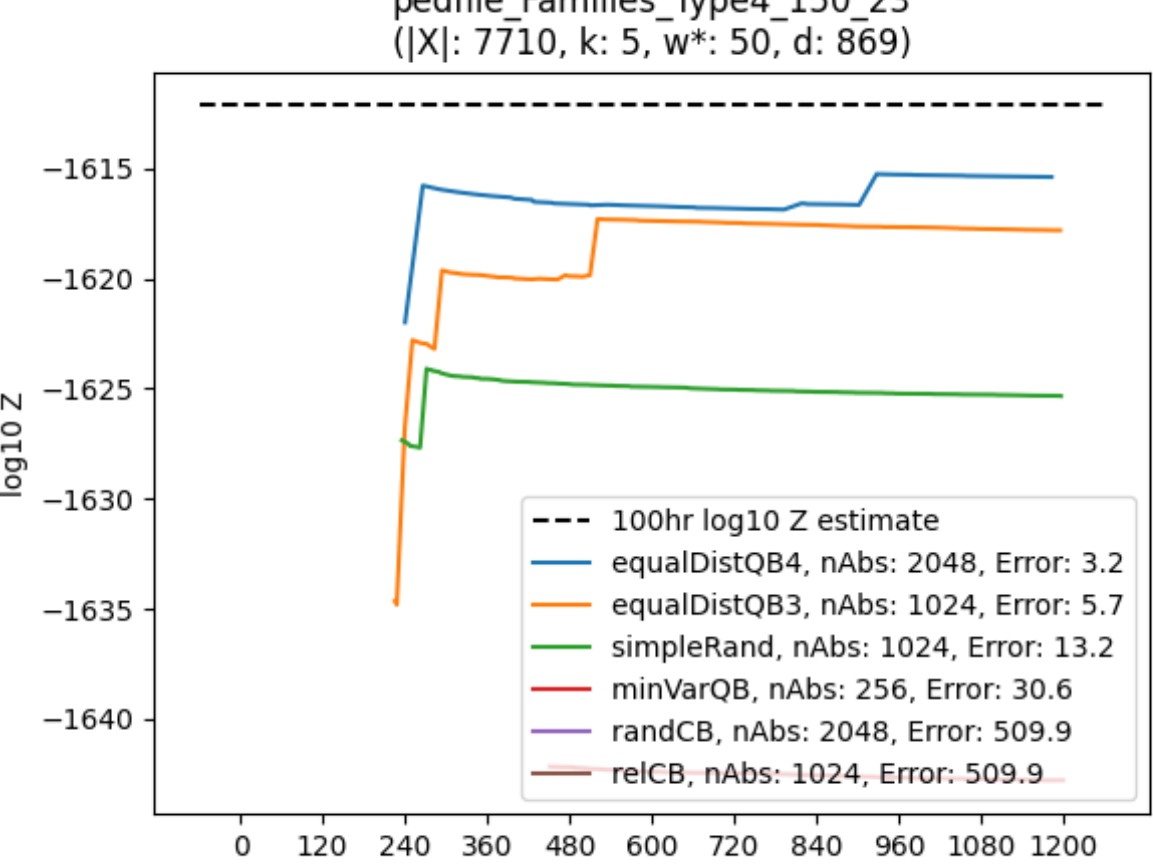

**Plot 3:** Z estimates from various algorithms versus time on Linkage-Type4 problem grid20x20.f15 using $iB = 10$. The dashed black line shows the estimated true Z value.

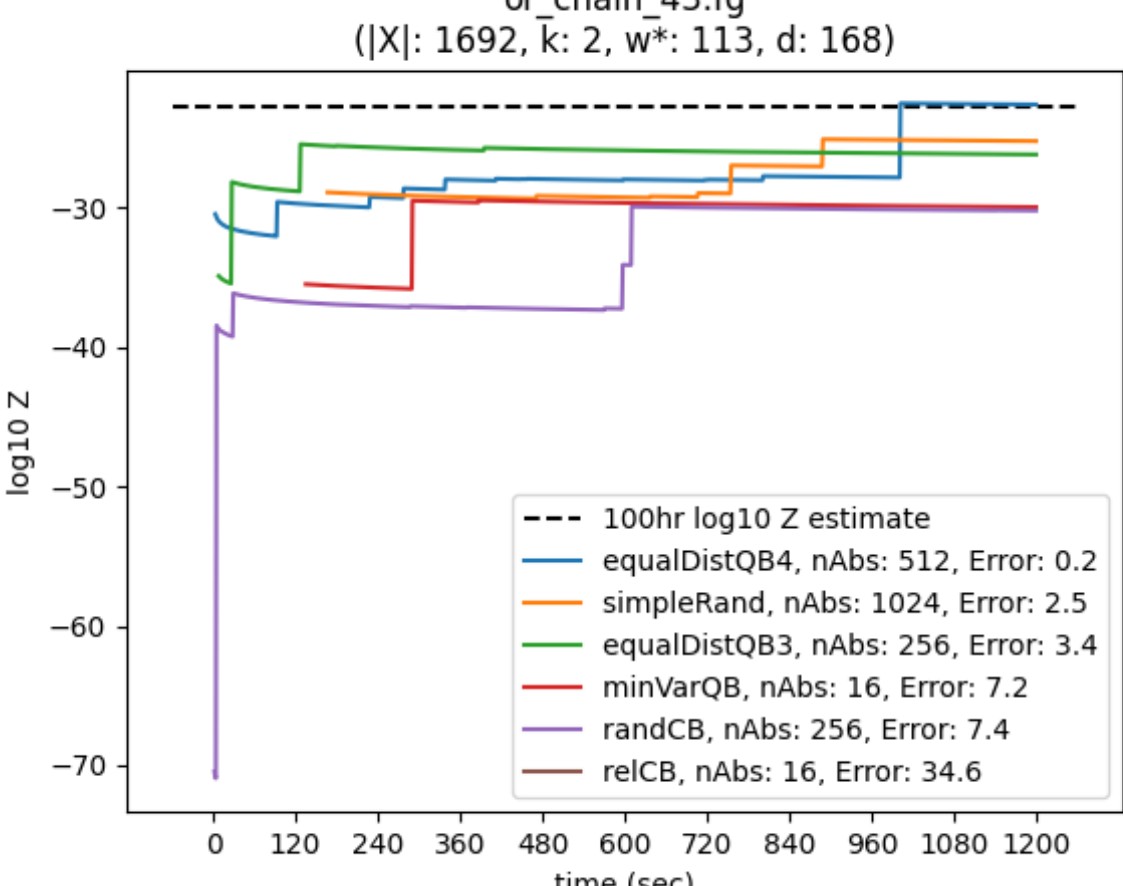

**Plot 4:** Z estimates from various algorithms versus time on Promedas problem or_chain_43.fg using $iB = 10$. The dashed black line shows the estimated true Z value.

# 8 ADDITIONAL RESULTS

## 8.1 PROBE SIZE

In our running abstraction example discussed in Supplemental Section 6, we observed that despite employing the same granularity, certain Ordered Partitioning Schemes may underutilize the allotted number of abstract states. Moreover, paths extended during initial iterations may become incomplete in subsequent iterations . These truncated paths may be pruned altogether and cut the number of nodes. To assess how effectively different schemes handle continual extension of paths, we fixed $nAbs$ at 2048 and plotted the Probe Size against the number of variables for each problem in the Promedas benchmark (Plot 5).

**Plot 5:** For the given abstraction granularity and benchmark, the size of the probe (in log10) relative to the number of problem variables (in log10) using iB-10.

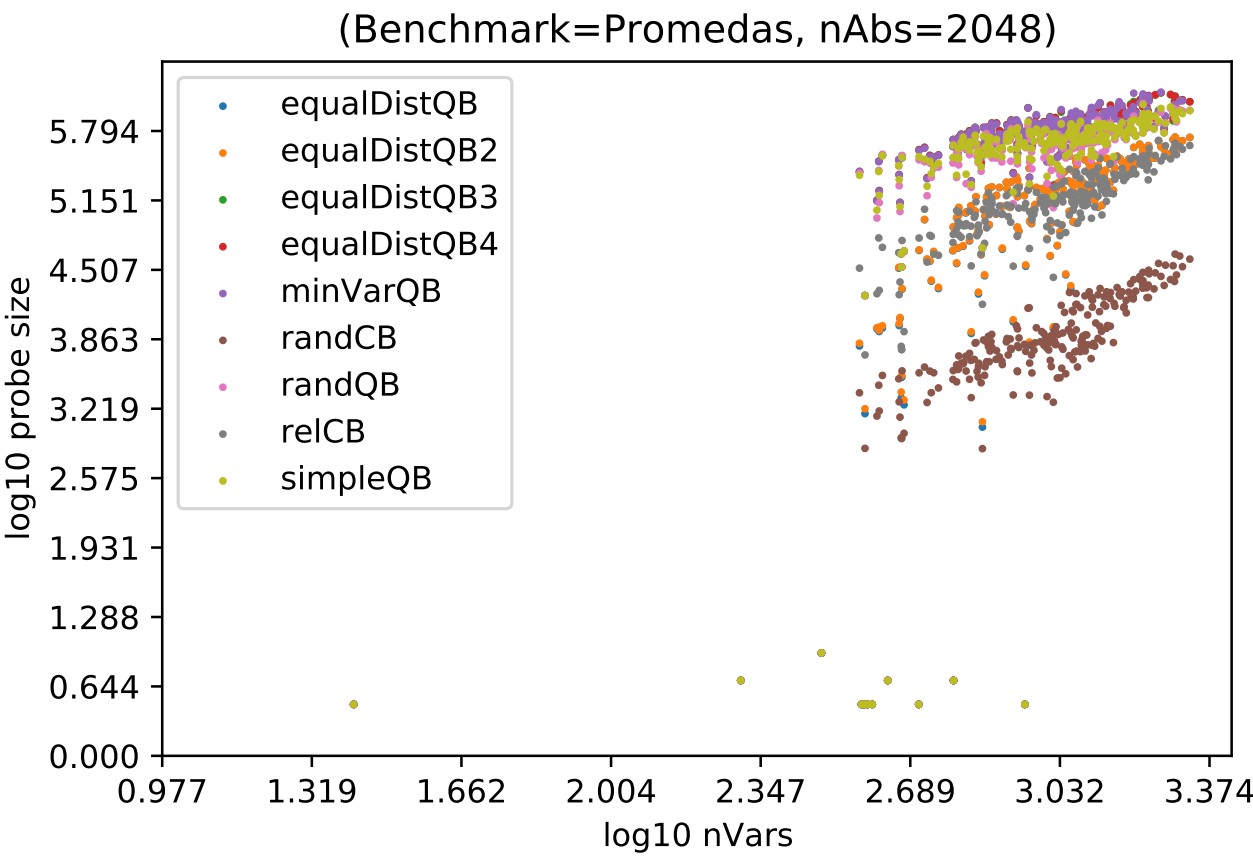

Even with the same granularity different abstraction functions can lead to vastly different utilization of abstract states, pruning, and thus probe sizes. Plot 5 highlights this. As seen in the plot (and generalizes across the different benchmarks and abstraction value classes) the simpleQB, minVarQB, equalDistQB3, equalDistQB4, and randQB schemes tend to produce larger probes, indicating more of the allotted abstract states utilized and fewer branches being pruned.

## 8.2 ABSTRACTION SPEED

In order to understand more about the speed of each scheme at performing abstractions, in Figure **??** we plot the number of samples versus average probe size for problems of the Promedas benchmark. (For other benchmarks and $nAbs$, please see

the Supplemental Materials).

**Plot 6:** For the given abstraction granularity and benchmark, the number of samples (in log10) relative to the probe size (in log10) using iB-10.

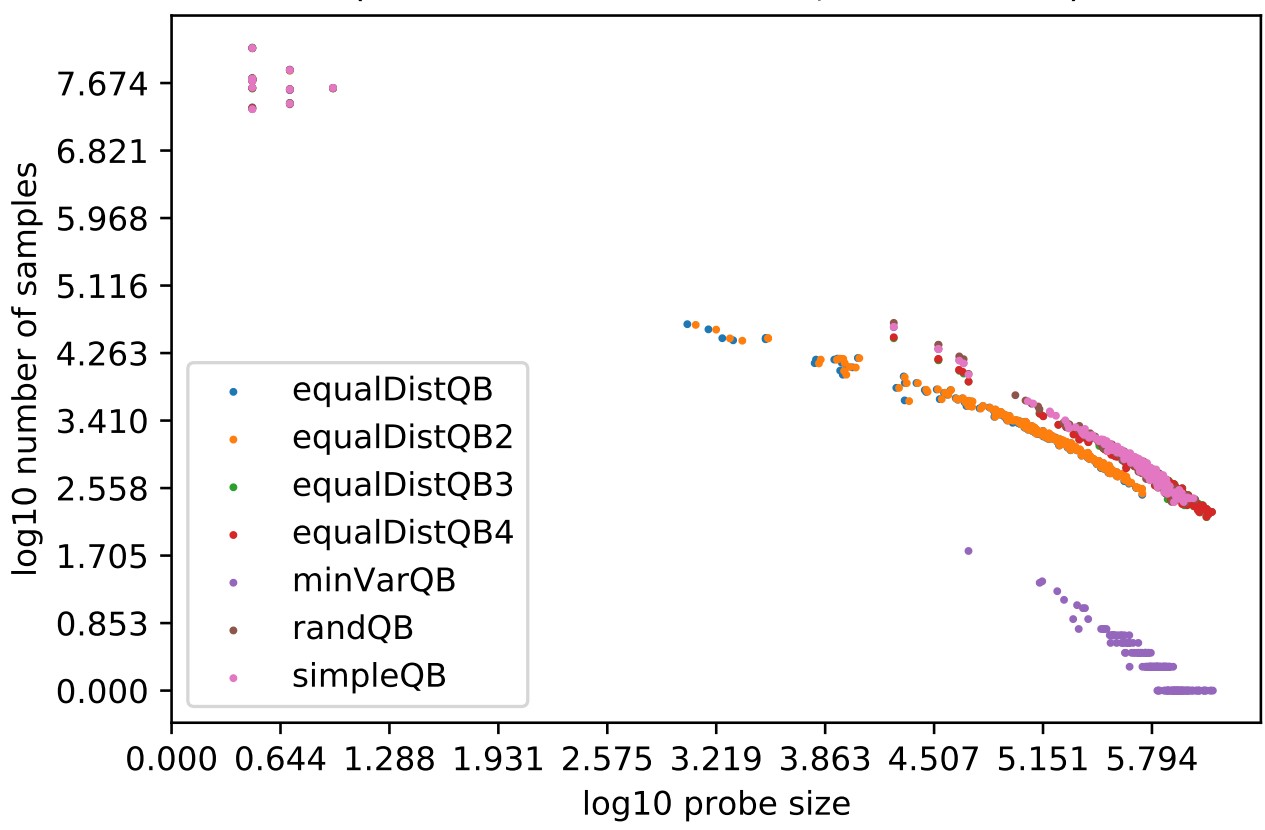

Plot 6 shows the number of samples that were able to be drawn relative to the size of generated probes, thus providing an understanding of the speed abstractions occur. As expected, we notice the minVar scheme (which utilizes a computationally intensive hierarchical clustering process to abstract nodes) has the lowest sample efficiency. The other schemes have comparable abstraction speeds.

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
