# OpenReview forum: "Value-Based Abstraction Functions for Abstraction Sampling"
_auai.org/UAI/2024/Conference — UAI 2024 poster_

### Official Review · Reviewer_o2y3 · 2024-03-17

**Q2-1 Originality-Novelty:** 3
**Q2-2 Correctness-Technical Quality:** 3
**Q2-5 Clarity Of Writing:** 4

**Q1 Summary And Contributions:**

The paper studies different abstraction functions to be used in the ASOS algorithm, which is sampling based algorithm for graphical model inference. In particular, author proposed a new framework of designing abstraction function for ASOS, which results in 21 different variations. Author empirically examined the effectiveness of the abstraction functions, and demonstrated that improved inference quaintly compared to abstraction function used in previous work.

**Q2-3 Extent To Which Claims Are Supported By Evidence:**

3: Good: the main claims are supported by convincing evidence (in the form of adequate experimental evaluation, proofs, (pseudo-)code, references, assumptions).

**Q2-4 Reproducibility:**

4: Excellent: key resources (e.g. proofs, code, data) are available and key details (e.g. proof sketches, experimental setup) are comprehensively described for competent researchers to confidently and easily reproduce the main results.

**Q3 Main Strengths:**

The paper proposed a framework of generating abstraction function for ASOS algorithm. In addition, author extensively studies different choice of the abstraction functions generated by this framework. This provides good reference of designing abstraction function in the future.

Author demonstrated that better abstraction functions can be generated using this framework compared to the existing abstraction function used in the existing literature.

In the experiment, author further demonstrated competitive performance of using a random abstraction function, which does not seem to be considered in the previous work.

**Q4 Main Weakness:**

Although the value-based abstraction framework seems to be very simple and easy to design new abstraction function, the framework does not seem to be general enough to capture existing abstraction functions. For example, IIUC, we cannot re-create the context based abstraction functions using the value-based abstraction framework.

**Q5 Detailed Comments To The Authors:**

Can author provide insights why does the ASOS estimate always under-estimates based on Plot 3 and 4 on the grids and or_chain benchmark? Does that imply bias on the sampling algorithm when the sample size is small?

**Q9 Complying With Reviewing Instructions:**

Yes

---

> ### Author Rebuttal · Authors · 2024-04-04
>
> Thank you very much for your review and questions.
>
> Yes, our newly proposed abstraction functions are not meant to generalize the previous CTX-based abstractions directly.  However, our goal is to incorporate more useful information into the abstractions – the context is a function only of the graph structure, while the heuristic function should have significantly more information about the similarity of different configurations.  We believe this class of abstractions is more powerful, but agree that it is possible to explore extensions of context-only abstractions as well.
>
> Underestimation: our method, like any importance sampling method, is an unbiased estimate of Z; but in typical practice most samples slightly underestimate, while a few (rare) samples overestimate by a larger amount. This is a common observation about importance sampling generally, due to the proposal distribution’s tails.

---

### Official Review · Reviewer_h1DL · 2024-03-17

**Q2-1 Originality-Novelty:** 2
**Q2-2 Correctness-Technical Quality:** 3
**Q2-5 Clarity Of Writing:** 3

**Q1 Summary And Contributions:**

Abstraction Sampling (AS) enhances Importance Sampling by using abstractions: grouping similar sampled nodes into abstract states.

This paper provides a detailed study of new abstraction function schemes for AS algorithms over AND/OR search graphs. The study introduces 3 new classes of abstraction functions, each combined with 7 partitioning strategies.

The paper empirically identifies three well-performing configurations.

**Q2-3 Extent To Which Claims Are Supported By Evidence:**

3: Good: the main claims are supported by convincing evidence (in the form of adequate experimental evaluation, proofs, (pseudo-)code, references, assumptions).

**Q2-4 Reproducibility:**

3: Good: key resources (e.g. proofs, code, data) are available and key details (e.g. proofs, experimental setup) are sufficiently well-described for competent researchers to confidently reproduce the main results.

**Q3 Main Strengths:**

The paper proposes several novel configurations and compares against current heuristics, showing improved performance.
The experimental setup is in line with previous study (e.g., Kask et al.).

**Q4 Main Weakness:**

I estimate the impact to be relatively minor.

~The evaluation metric (average error instead of average relative error) seems flawed to me.~

Edit: author's reply addressed my concerns regarding the evaluation metric.

**Q5 Detailed Comments To The Authors:**

Does the average error make sense? I.e., unless Z is comparable in size within an instance class (DBN, Grids,...), should we not consider the average **relative** error instead??


The intuition behind the Heuristic and Ancestral Branching-Based Abstraction is not clear to me. The explanation suggest $hr(n)$ as a similarity measure. But if $h(n)$ and $r(n)$ are estimates of $Z(n)$ and $R(n)$ respectively, then $hr(n)$ is a measure of how close the estimates are, and is at best equal to 1? I do not see a link with estimating the similarity between nodes.


Would it be possible to combine different heuristics, i.e., decide after an expansion which heuristic to use, could that work?


I appreciated the running example for the partitioning schemes, and in the appendix the example of a sampling trace. What was not clear to me however: does the sampling approach as a whole generate multiple probes, or just one probe of multiple paths? While reading I felt like I was missing the complete picture.



The paragraph explaining $R(n)$ is not very clear. In particular $br(n_X)$, from the explanation it is not clear to me that $B$ (Figure 2) would be in $br(n_X)$.



Several typos:

* stochasticallh
* In Def 4.1, $nasb$ instead of $nAbs$
* oreder
* Theorem 4.1 is confusing. Is the point after $c_i$ supposed to be a comma, or is the following "whenever" supposed to be capitalized? Part of my confusion comes from $c_i > 0$ followed by the case $h(n)r(n) = 0$ that implies $c_i = 0$.
* Background example, should $Z(n_{A=0,B})$ not be $Z(B_{n_{A=0}})$?
* varaint
* aaims
* abstarction
* the the
* Table 3 show RAND ... -> shows

**Q9 Complying With Reviewing Instructions:**

Yes

---

> ### Author Rebuttal · Authors · 2024-04-04
>
> Thank you very much for your review and questions.
>
> In fact, we use the absolute error on log(Z), which corresponds to the log-ratio of the true Z and the Z-estimate, closely related to the relative error on Z. This is typical of most work estimating the partition function or probability of evidence.  (It would be unusual to look at relative error on log(Z), since log(Z) may be exactly zero, for example in a normalized Bayesian network).
>
> Regarding impact: estimating the partition function is a classic and difficult task, and importance sampling a workhorse of Monte Carlo estimators, particularly in discrete models where modern MCMC methods are difficult to apply.  Our proposed schemes effectively dominate the prior state-of-the-art.  Tables 3a & b [pg. 7] and Figure 3 [pg. 8] show that RAND, equalDistQB3, and equalDistQB4 outperform CTX on every benchmark, often by a lot.  (For comparison, in [Broka et al. 2018] and [Kask et al. 2022] CTX outperformed a number of existing approaches).   While we study performance on the partition function, as a general Monte Carlo approach our scheme can be modified for other tasks such as to estimate conditional expectations, etc.
>
> HR-scheme: Theorem 4.1 [pg. 4] shows us that if $\frac{h(n)r(n)}{Z(n)R(n)}$ is constant over $n$ for nodes abstracted together (we will call this “identically accurate”), the resulting probe’s estimate is exact (has zero variance).  However, without access to $Z(n)$ or $R(n)$ we cannot evaluate this directly.  The intuition of HR is that grouping based on the value of $h(n) r(n)$ may also result in sets of nodes with similar $Z(n)R(n)$, and thus close to “identically accurate”.
>
> Each probe of abstraction sampling corresponds to a subtree of the full search tree, and so usually includes multiple configurations (assignments to all variables). Then, we generate many such probes and use them, and their weights, to provide a Monte Carlo estimate.  We will try to ensure this is clear in the final version; please feel free to follow up here in the meantime.
>
> Regarding interleaving or mixing heuristic or abstraction methods within a probe: yes, it is certainly possible to do so while retaining unbiasedness. The key would be to do so strategically.  It is definitely an interesting idea to select an abstraction method based on some property of the expansion; for example, the wMBE heuristic is often less accurate early in the probes, so one could use RAND for the higher levels and then switch to equalDistQB to provide abstractions at lower levels.
>
> Regarding $R(n_{A=0,C=1})$ and $br(n_{A=0,C=1})$ in Figure 1 [pg. 2]:  To find $br(n_{A=0,C=1})$, we look for all ancestor AND nodes (rectangles) of $n_{A=0,C=1}$ that have multiple children, in this case $br(n_{A=0,C=1}) = \set{n_{A=0}}$.  The purpose of the ancestor branching mass $R(n_{A=0,C=1})$ is to capture the contribution of assignments to variables not on the path to $n_{A=0,C=1}$.  In our final submission we will describe this more thoroughly and include an additional background-focused supplemental document to further explain such underlying concepts with examples to improve accessibility.

---

### Official Review · Reviewer_MdZq · 2024-03-20

**Q2-1 Originality-Novelty:** 2
**Q2-2 Correctness-Technical Quality:** 3
**Q2-5 Clarity Of Writing:** 4

**Q10 Ethical Concerns:**

The proposed work does not raise any new potential concerns other than the ones that are automatically present when using real-world (potentially sensitive) data.

**Q1 Summary And Contributions:**

This paper proposes a new method to estimate the partition function for probabilistic graphical models. More specifically, they propose new abstraction functions in the context of abstraction sampling and perform experiments to show that the newly proposed methods outperform the state-of-the-art techniques (i.e. context-based sampling schemes). The new abstraction functions are formed by combining a value function (one of HR, HRB or QB) with a partitioning scheme (one of seven schemes), consequently yielding 21 new abstraction functions. Along with that they also introduce a randomized abstraction function that seems to fares comparably.

**Q2-3 Extent To Which Claims Are Supported By Evidence:**

3: Good: the main claims are supported by convincing evidence (in the form of adequate experimental evaluation, proofs, (pseudo-)code, references, assumptions).

**Q2-4 Reproducibility:**

2: Fair: key resources (e.g. proofs, code, data) are unavailable but key details (e.g. proof sketches, experimental setup) are sufficiently well-described for an expert to confidently reproduce the main results.

**Q3 Main Strengths:**

- Writing is very clear and lucid at least in the initial sections.
- Efficient use of (running) examples where necessary.
- Experiments seem rigorous, thorough and convincing.

**Q4 Main Weakness:**

- Contributed abstraction functions don't seem overly complicated to devise.
- The fact that the extremely simple randomized abstraction function also performs similarly to the best among the remaining abstraction functions seems to undermine the necessity of the 21 abstraction functions.
- Could potentially increase the benchmark size used for the experiments.
- Certain parts of the "General Background" felt rushed/hand-wavy.

**Q5 Detailed Comments To The Authors:**

(mostly minor remarks and typos)

- [p2, right column, 1st line] the notion of "path to $n$ that fully instantiates all $X' \in \alpha$" is a bit unclear to me
- [p2, right column, 4rd para] how is $Z(n)$ defined for a leaf node?
- [p2, right column, 5th para] "... there may *be* an intermediate node ..."
- [p3, left column, middle of 2nd para] (typo) "levl" -> "level"
- [p3, left column, 3rd para] "Algorithm 1 provides a high-level description" of what? (please specify "high-level description of AOAS" or something like that)
- [p3, right column, Algorithm 1, Step 2(b)(ii)] (typo) "stochasticallh" -> "stochastically"
- [p3, right column, last line] missing hyphen "order-consistent manner" (also remove spurious full stop that comes immediately after (on the next page))
- [p4, left column, Definition 4.1] (typo) "oreder consistent" -> "order-consistent"
- [p4, left column, Definition 4.1] (typo) "$nasb$" -> "$nAbs$", also add hyphen between "$nAbs$-allowed"
- [p4, left column, Definition 4.1] parenthesis mismatch in _"($n_1 \in A_i$ and $n_2 \in A_j$ where $i < J) \iff \mu(n_1) \leq \mu(n_2))$"_
- [p4, left column, 3rd para] the acronym "wMBE" is only defined in the experimental section
- [p4, right column] all the partitioning schemes here have the suffix "VB" but the suffix disappears in the experimental section, please be consistent
- [p4, right column, SimpleVB] (rephrase) "... nodes are partitioned into *abstract states with* [approximately] equal cardinality"
- [p5, left column, 2nd para] (typo) "can stillyields"
- [p5, left column, 5th para] (typo) "aaims" -> "aims"
- [p5, right column, 2nd para] how is the pseudotree generated for the experiments?
- [p5, right column, 3rd para] "... Pedigree, and Promedas used by [Kask et al., 2020]" use `\citet` or `\textcite` here instead
- [p6, left column, 2nd para] "performed well across all benchmarks", could you please be more precise and specify quantitatively what it really means?
- [p8, conclusion] (typo) "abstarction" -> "abstraction"
- [p8, conclusion] (typo) "one of the best scheme ..." -> "one of the best schemes ..."

**Q9 Complying With Reviewing Instructions:**

Yes

---

> ### Author Rebuttal · Authors · 2024-04-04
>
> Thank you very much for reading and reviewing our paper, and for your detailed location of typos.
>
> While our various proposed abstraction functions are not particularly complex to define, each is founded on different principles and a priori it is not obvious which, if any, might perform well.  For example, earlier work had suggested heuristic based abstraction functions, but abandoned this line of work after inconclusive results.  By explicitly delineating these various intuitions and exploring many instances, we are pleased we have identified new abstraction functions that substantially outperform prior state-of-the-art. We also try to provide comprehensive comparisons, enabling others to modify or extend these approaches and easily compare performance.
>
> Yes, one of our findings is the surprising robustness of the RAND scheme; but this also serves to identify a few techniques (equalDistQB3 and equalDistQB4) that are stronger overall given a good heuristic.
>
> If by benchmark size you are referring to the size of each problem: yes we are interested in and do give results on large benchmarks (see Table 3b [pg 7]). However, in large problems the true Z value is unknown.  In Kask et al [2020] the target Z was estimated using a much longer sampling run; but even a few orders of magnitude more samples can be insufficient to overcome a worse sampling distribution, so some care must be taken.  We decided to focus mainly on problems with known answers, but using weakened heuristics and shorter run times to increase difficulty.  We do also have additional results using longer runs with the new schemes to estimate the true answer to the large problems, and can include these in the final submission.
>
> If by benchmark size you are referring to the number of problems: our evaluation is on over 480 widely used benchmark problems.  Nevertheless, we have also tested on UAI 2022 competition problems for which the new schemes again substantially outperform the CTX based schemes.
>
> Regarding the background: given the submission page limit, we devoted more space to contributions and results; however, the final (extended) version allows two additional pages, and we will expand the background to make it more broadly accessible.  We will also include an additional background-focused supplemental document to further explain underlying concepts and provide examples to improve accessibility.  We hope these changes will address your concerns.

---

### Official Review · Reviewer_9KNF · 2024-03-20

**Q2-1 Originality-Novelty:** 3
**Q2-2 Correctness-Technical Quality:** 3
**Q2-5 Clarity Of Writing:** 3

**Q10 Ethical Concerns:**

No.

**Q1 Summary And Contributions:**

The paper proposes 22 new abstraction functions for abstraction sampling--a Monte Carlo estimation method for the partition function of a graphical model. Via an extensive experimental analysis, some of the suggested abstraction functions are shown to consistently outperform the state of the art.

**Q2-3 Extent To Which Claims Are Supported By Evidence:**

4: Excellent: all claims are supported by very convincing evidence (in the form of comprehensive experimental evaluation, rigorous mathematical proofs, detailed (pseudo-)code, precise references, well-motivated and realistic assumptions) and the authors deliver what they promise.

**Q2-4 Reproducibility:**

2: Fair: key resources (e.g. proofs, code, data) are unavailable but key details (e.g. proof sketches, experimental setup) are sufficiently well-described for an expert to confidently reproduce the main results.

**Q3 Main Strengths:**

The paper proposes many new abstraction functions and investigates them via a very detailed experimental study. The technical details appear to be sound.

**Q4 Main Weakness:**

Although I think the overall clarity of writing is good enough, the paper is quite dense, with many useful explanations (including more extensive preliminaries and details about the proposed abstraction functions) relegated to the supplementary material. Perhaps some of that material could be moved into the main paper at the expense of a few plots/tables describing experimental results.

The experimental setup is described well, but the implementation is not included in the supplementary material.

**Q5 Detailed Comments To The Authors:**

* How would you interpret the fact that RAND is performing so well?
* In the description of equalDistVB, mu(n) should be mu(n').
* "N" is boldface in the captions of the tables but not boldface in the tables themselves.
* There are many typos such as "that that", "oreder", "nasb", "closed" (instead of "close"), "it inspire", "with-in", "stillyields", "varaint", "From Table 5 that", and "abstarction".

**Q9 Complying With Reviewing Instructions:**

Yes

---

> ### Author Rebuttal · Authors · 2024-04-04
>
> Thank you very much for reading our paper and providing your review.
>
> Yes, we also agree the writing is quite dense in this initial submission.  In the final extended draft (which allows for two additional pages) we will expand the background and move more of the preliminaries and details of the abstraction functions from the supplemental to the main paper as you suggest.  We will also include an additional background-focused supplemental document to further explain underlying concepts and provide examples to make the paper more accessible.
>
> We will gladly include implementation of our abstraction functions in the final submission.
>
> Regarding the completely random (RAND) scheme’s strong performance: while it may seem surprising, the randomness in the stratification provided by RAND can allow for a diverse set of probes which can help generate configurations that are low probability (under the proposal), but may be relevant to the target distribution.  Thus, RAND can provide a surprisingly useful balance between sampling configurations that are likely under the proposal, while occasionally forcing the sampling of less probable configurations.

---

### Official Review · Reviewer_KQNe · 2024-03-23

**Q2-1 Originality-Novelty:** 3
**Q2-2 Correctness-Technical Quality:** 3
**Q2-5 Clarity Of Writing:** 2

**Q1 Summary And Contributions:**

The paper studies the abstraction functions that guide stratification in the abstraction sampling. The authors propose three new classes of abstraction functions and seven distinct partitioning schemes, making up 21 new abstraction functions. Extensive experiments are conducted to test the performance of the new proposed 21 new abstraction functions.

**Q2-3 Extent To Which Claims Are Supported By Evidence:**

3: Good: the main claims are supported by convincing evidence (in the form of adequate experimental evaluation, proofs, (pseudo-)code, references, assumptions).

**Q2-4 Reproducibility:**

3: Good: key resources (e.g. proofs, code, data) are available and key details (e.g. proofs, experimental setup) are sufficiently well-described for competent researchers to confidently reproduce the main results.

**Q3 Main Strengths:**

1. The paper proposes several novel value-based abstraction classes and partition schemes. And some of the newly proposed abstraction functions obtain great empirical performance;
2. The performance of each abstraction function is fully tested using a wide range of experiments;
2. The figures and tables in the paper are beautiful and clear

**Q4 Main Weakness:**

The main problem with the paper lies in the clarity of the writing.
1. There are so many minor grammar problems (like "but it inspire some related expressions", "be the child OR node of $n_Y$ that that is also on the path to X");
2. Some paragraphs and sections are hard to follow, like the content after the Notation paragraph in section 2, which will be much more straightforward for readers unfamiliar with AND/OR graphs if several explanations of what these functions mean and several more examples can be added.

The paper will be much better after careful polishing.

**Q5 Detailed Comments To The Authors:**

Please see the weakness above.

**Q9 Complying With Reviewing Instructions:**

Yes

---

> ### Author Rebuttal · Authors · 2024-04-04
>
> Thank you very much for reading our paper and providing your review.
>
> Regarding typos and grammar mistakes, yes these will be corrected and we apologize for any confusion.
>
> We will also expand the background in the final extended version (which allows for two additional pages) to make the paper more accessible.  Furthermore, we will include an additional extended background-focused supplemental document to explain underlying concepts and provide examples.  We hope this can help bridge the gap you mention.

---

### Meta-Review · Area_Chair_QSV8 · 2024-04-17

Abstraction sampling is one Monte Carlo integration method for estimating the partition function of a graphical model. The paper presents a few new abstraction functions and conducts an extensive experimental analysis against various different alternative choices.

The reviewers generally found the empirical study convincing and the results interesting and useful, compensating for the lack of truly novel technical ideas. The writing of the paper was considered somewhat dense, yet otherwise clear and of good quality.